# On the Representation of Solutions to Elliptic PDEs in Barron Spaces

**Ziang Chen**
Department of Mathematics
Duke University
Box 90320, Durham, NC 27708
`ziang@math.duke.edu`

**Jianfeng Lu**
Departments of Mathematics, Physics, and Chemistry
Duke University
Box 90320, Durham, NC 27708
`jianfeng@math.duke.edu`

**Yulong Lu**
Department of Mathematics and Statistics
Lederle Graduate Research Tower
University of Massachusetts
710 N. Pleasant Street, Amherst, MA 01003
`lu@math.umass.edu`

## Abstract

Numerical solutions to high-dimensional partial differential equations (PDEs) based on neural networks have seen exciting developments. This paper derives complexity estimates of the solutions of $d$-dimensional second-order elliptic PDEs in the Barron space, that is a set of functions admitting the integral of certain parametric ridge function against a probability measure on the parameters. We prove under some appropriate assumptions that if the coefficients and the source term of the elliptic PDE lie in Barron spaces, then the solution of the PDE is $\epsilon$-close with respect to the $H^1$ norm to a Barron function. Moreover, we prove dimension-explicit bounds for the Barron norm of this approximate solution, depending at most polynomially on the dimension $d$ of the PDE. As a direct consequence of the complexity estimates, the solution of the PDE can be approximated on any bounded domain by a two-layer neural network with respect to the $H^1$ norm with a dimension-explicit convergence rate.

## 1 Introduction

Inspired by the tremendous success of deep learning in diverse machine learning tasks including image classification, natural language processing, and artificial intelligence, there has been growing interest in exploring scientific and engineering applications of deep learning [36, 32, 34, 26, 47]. As partial differential equations (PDEs) play a fundamental role in almost all branches of sciences and engineering, numerical solutions to PDE problems based on neural networks have become an important research direction in scientific machine learning [25, 6, 23, 17, 7, 10, 22]. Among the various directions, numerical solutions to high-dimensional PDEs – the unknown function depending on many variables – are perhaps the most exciting possibility, as solving such PDEs has been a long-

35th Conference on Neural Information Processing Systems (NeurIPS 2021).

standing challenge and breakthrough would lead to tremendous progress in fields such as many-body physics [4, 11, 18], multiple agent control [35, 17], just to name a few.

Numerical solutions to low-dimensional PDEs, such as Navier-Stokes equation in fluid dynamics, has become a standard practice after decades of work. However, the computational cost of the conventional numerical methods for PDEs grows exponentially with the dimension, as a manifestation of the curse of dimensionality (CoD). Given a target accuracy $\epsilon$, conventional methods, such as finite element or finite difference, would need a mesh size of $\mathcal{O}(\epsilon)$, and thus degree of freedom on the order of $\mathcal{O}(\epsilon^{-d})$, where $d$ is the dimension of the problem. Such complexity severely limits the numerical solutions to PDEs in high dimension, such as the many-body Schrödinger equations from quantum mechanics and the high-dimensional Hamilton-Jacobi-Bellman equations from control theory. Neural networks, in particular deep neural networks, provide a promising way to overcome the CoD in representing functions in high dimension. It is thus a natural idea to parametrize the solution ansatz to a PDE as neural networks and to employ variational search for the optimal parameters. Various neural network methods [25, 6, 7, 10, 33, 17, 41, 46, 16, 5] for PDEs have been proposed recently and some of them have demonstrated great empirical success in solving PDEs of hundreds and thousands of dimensions [7, 10, 17], much beyond the capability of conventional approaches. Question remains though on theoretical analysis of such neural-network based methods for solving high-dimensional PDEs. While there have been some recent progress on approaches including physics-informed neural networks [37, 31, 38] and the deep Ritz method [28, 27], many questions still remain open. Among them, a fundamental question is

*Whether the solution of a high-dimensional PDE can be efficiently approximated by a neural network, and if so, how to quantify the complexity of the neural network representation with respect to the increasing dimension?*

**Our contributions**   The focus of the current study takes a functional-analytic approach to this question. Namely, we identify a function class suitable for neural network approximations and prove that the solutions to a class of PDEs can be well approximated by functions in this class. More specifically, the PDE we consider is a family of second-order elliptic PDEs of the form

$$\mathcal{L}u = -\nabla \cdot (A\nabla u) + cu = f \text{ on } \mathbb{R}^d. \tag{1.1}$$

We choose to work with the Barron class of functions defined in [8] (see also [1]), which is a class of functions admitting the integral of certain parametric ridge function against a probability measure on the parameters; see Definition 2.2 for a precise description. This Barron space is inspired by the pioneering work by Barron [2], where he proved that a class of functions whose Fourier transform has the first order moment can be approximated by two-layer networks without CoD. The main result of our work, stated informally, is the following; a more precise statement can be found in Section 2.3.

**Main Theorem (informal version)**   If the coefficients $A, c$ and the source term $f$ of the second-order elliptic PDE (1.1) are all Barron functions, then the solution $u^*$ can be approximated by another Barron function $u$ such that $\|u - u^*\|_{H^1} \le \epsilon$, where the Barron norm of $u$ is upper bounded by $\mathcal{O}((d/\epsilon)^{C \log(1/\epsilon)})$. Moreover, if the Barron space is defined by the cosine activation function, then the upper bound on the Barron norm can be improved to $\mathcal{O}(d^{C \log 1/\epsilon})$.

We note that while the better rate is only obtained for the cosine activation function, such periodic activation function has indeed been found effective in certain PDE related tasks, see e.g., [42].

Since the Barron functions can be approximated on a finite domain $\Omega$ w.r.t. $H^1$ norm by two-layer neural networks with a rate $\mathcal{O}(1/\sqrt{k})$ where $k$ is the network width (see Theorem 2.5), the theorem above directly implies that there exists a two-layer network $u_k$ with the number of widths $k = \mathcal{O}((d/\epsilon)^{C \log(1/\epsilon)})$, or $k = \mathcal{O}(d^{C \log 1/\epsilon})$ if the activation function is cosine, such that $\|u_k - u^*\|_{H^1(\Omega)} \le \epsilon$. Therefore in our setting the solution can be approximated by a two-layer neural network without CoD, namely the complexity depends at most polynomially on the dimension $d$ for fixed $\epsilon$. Alternatively, we can rewrite the rates as $\mathcal{O}((1/\epsilon)^{C(\log d + \log 1/\epsilon)})$ and $\mathcal{O}((1/\epsilon)^{C \log d})$ to contrast with that of conventional grid-based numerical methods for PDEs, which scales as $\mathcal{O}((1/\epsilon)^d)$. We observe that the dependence on $d$ is replaced with $\log d$ in the complexity bound for neural network approximations.

We emphasize that such approximation result does not follow directly from the universal approximation property of neural networks for Barron functions since it is not a priori known that the solution to

the PDE is a Barron function. In fact, directly imposing regularity or complexity assumption on the solution itself is unreasonable since the solution is unknown and its fine properties are generally inaccessible. Our main contribution is to establish the fact that the solution can be indeed approximated by a Barron function, under the assumption that coefficients and the right hand term of the PDE are Barron. From a mathematical point of view, our main theorem is in the same spirit as regularity estimates of PDEs, which are of crucial importance in the study of PDEs. While such regularity estimates are well developed in low dimension, the extension to results in high dimension is highly non-trivial and is the main focus of our work.

**Related works**  Several theoretical work have been devoted to the above representation question. It has been established in [15, 20, 14] that deep neural networks can approximate solutions to certain class of parabolic equations and Poisson equation without CoD. The major limitation of those work lies in that the PDEs considered in those work must admit certain stochastic representation such as the Feymann-Kac formula and it seems difficult to generalize the proof techniques to broader classes of PDEs with no probabilistic interpretation. The work [28, 27] analyzed a priori generalization error of two-layer networks for solving elliptic PDEs and the Schrödinger eigenvalue problem on a bounded domain with Neumann boundary condition by assuming that the exact solutions lie in certain spectral Barron space, where the later was rigorously justified with a new regularity theory of the PDE solutions in the spectral Barron space. Similar generalization analysis was carried out in [29] for second-order PDEs and in [19] for general even-order elliptic PDEs, but without justifying the Barron assumption on the solution. Compared to those work, our work focuses on deriving complexity estimates of the solution in the integral-representation-based Barron space, which is more flexible and arguably more suitable for high-dimensional settings, see e.g., discussion in [8]. The work [9] established such estimates in the Barron space for certain specific PDEs that essentially admit explicit solution, whereas we aim to prove such estimates for general elliptic PDEs for which the analytical ansatz is not available. The work [30] is closest to ours where the authors proved that the solution of the same type of elliptic PDE with a Dirichlet boundary condition can be approximated by a (deep) neural networks with at most $\mathcal{O}(poly(d)N)$ parameters if the coefficients of the PDE are approximable by neural networks with at most $N$ parameters. While our overall approach based on iterative scheme borrows idea from [30], our result differs and improves theirs in many aspects: (1) Our result shows that the solution can be well approximated without CoD by a two-layer neural network with a single activation whereas the result in [30] requires a deep network which uses a mixture of at least two activation functions; (2) Our PDE is set up on the whole space rather than a compact domain, so our setting covers some important PDEs in physics, such as the stationary Schrödinger equation; (3) The result in [30] relies on another key assumption that the source term lies within the span of finitely many eigenfunctions of the elliptic operator whereas our result completely removes such assumption. This is achieved by utilizing a novel preconditioning technique to uniformly control the condition number of the iterative scheme that underpins the proof of our main theorem.

**Organization**  The rest of this paper will be organized as follows. In Section 2.1 we set up the PDE problem on the whole space and in Section 2.2 we introduce the definition of Barron functions and discuss their $H^1$-approximation by two-layer networks (see Theorem 2.5). Our main theorems are stated in Section 2.3. We present the sketch proofs of the main theorems in Section 3 and defer the complete proof to Appendix. The paper is concluded with discussions on some future directions.

## 2  Problem setup and main results

### 2.1  Problem description

**Notations**  Throughout this paper, we use $\|v\|$ to denote the Euclidean norm of a vector $v \in \mathbb{R}^d$. For a matrix $A \in \mathbb{R}^{d \times d}$, we denote its operator norm by $\|A\| = \sup_{v \in \mathbb{R}^d \setminus \{0\}} \frac{\|Av\|}{\|v\|}$. For $R > 0$, we denote by $\overline{B}_R^d$ the closed ball in $\mathbb{R}^d$ centered at 0 with radius $R$, i.e., $\overline{B}_R^d = \{x \in \mathbb{R}^d : \|x\| \le R\}$.

Recall that we consider the $d$-dimensional second-order elliptic PDE (1.1). To guarantee the existence and uniqueness of the weak solution in $H^1(\mathbb{R}^d)$, we make the following minimum assumptions on coefficients $A, c$ and right-hand side $f$; this assumption will be strengthened in our main representation theorem.

**Assumption 2.1.** $A(x) = (A_{ij}(x))_{1 \leq i,j \leq d}$ *is symmetric with* $\|A(x)\| \leq a_{\max} < \infty$ *and uniformly elliptic, that is for some* $a_{\min} > 0$, *it satisfies*

$$\xi^\top A(x)\xi \geq a_{\min} \|\xi\|^2, \quad \forall\, x, \xi \in \mathbb{R}^d.$$

*We also assume that* $0 < c_{\min} \leq c(x) \leq c_{\max} < \infty$ *and* $f \in L^2(\mathbb{R}^d)$.

Under Assumption 2.1, a standard argument using the Lax-Milgram theorem implies that there exists a unique weak solution $u^* \in H^1(\mathbb{R}^d)$, such that $\mathcal{L}u^* = f$ in $H^{-1}(\mathbb{R}^d)$ which is the dual space of $H^1(\mathbb{R}^d)$, i.e.,

$$\int_{\mathbb{R}^d} A\nabla u^* \cdot \nabla v dx + \int_{\mathbb{R}^d} cu^* v dx = \int_{\mathbb{R}^d} fv dx, \quad \forall v \in H^1(\mathbb{R}^d).$$

Our ultimate goal is to show that the solution can be approximated by a two-layer neural network on any bounded subset of $\mathbb{R}^d$ with respect to the $H^1$ norm with a rate scaling at most polynomially in the dimension. Notice that in general one cannot hope to obtain an approximation result on the whole space $\mathbb{R}^d$ because the asymptotic behavior of a neural network function (determined by the activation) at infinity may mismatch that of the target function $u^*$. On the other hand, it is well-known that the convergence rate of neural networks for approximating functions in standard Sobolev or Hölder spaces still suffers from the CoD [44, 45]. Therefore to obtain a rate without CoD for the neural networks approximation to the solution $u^*$, we need to argue that $u^*$ lies in a suitable smaller function space which has low complexity compared to Sobolev or Hölder spaces. We will work with the Barron space and show that $u^*$ is arbitrarily close to a Barron function which can be approximated by a two-layer neural network without CoD.

## 2.2 Barron spaces

The definition of Barron space is strongly motivated by the two-layer neural networks. Recall that a two-layer neural network with $k$ hidden neurons is a function of the form

$$u_k(x) = \frac{1}{k} \sum_{i=1}^{k} a_i \sigma(w_i^\top x + b_i), \quad x \in \mathbb{R}^d. \tag{2.1}$$

Here $\sigma : \mathbb{R} \to \mathbb{R}$ is some activation function and $(a_i, w_i, b_i) \in \mathbb{R} \times \mathbb{R}^d \times \mathbb{R}$, $i = 1, 2, \ldots, k$ are the network parameters. If the parameters are randomly chosen accordingly to some probability distribution, then in the infinite width limit the averaged sum in (2.1) formally converges to the following probability integral

$$u_\rho(x) := \int a\sigma(w^\top x + b)\rho(da, dw, db), \quad x \in \mathbb{R}^d, \tag{2.2}$$

where $\rho$ is a probability measure on the parameter space $\mathbb{R} \times \mathbb{R}^d \times \mathbb{R}$. Observe that (2.1) is a special instance of (2.2) if we take $\rho(a, w, b) = \frac{1}{k} \sum_{i=1}^{k} \delta(a - a_i, w - w_i, b - b_i)$.

The Barron norms and Barron spaces are then defined as follows, where we require the marginal measure in $w$ to have compact support. This is because that the (formal) first-order and second-order partial derivatives of $u_\rho(x)$ would involve with components of $w$ by chain rule. By adding some uniform bounds on $w$, we can to control the Barron norms after taking derivatives. In the subsequent discussion, we may also need to restrict our attention on functions defined on a bounded set. Therefore we present below the formal definition of a Barron function defined any domain $\Omega \subset \mathbb{R}^d$.

**Definition 2.2.** *Fix* $\Omega \subset \mathbb{R}^d$ *and* $R \in [0, +\infty]$. *For a function* $g = u_\rho$ *with some probability measure* $\rho$, *we define the Barron norm of* $g$ *on* $\Omega$ *with index* $p \in [1, +\infty]$ *and support radius* $R$ *by*

$$\|g\|_{\mathcal{B}_R^p(\Omega)} = \inf_\rho \left\{ \left( \int |a|^p \rho(da, dw, db) \right)^{1/p} : g = \int a\sigma(w^\top x + b)\rho(da, dw, db) \text{ on } \Omega, \right.$$

$$\left. \rho \text{ is supported on } \mathbb{R} \times \overline{B}_R^d \times \mathbb{R} \right\},$$

*where* $\overline{B}_R^d = \{x \in \mathbb{R}^d : \|x\| \leq R\}$. *The corresponding Barron space is then defined as*

$$\mathcal{B}_R^p(\Omega) = \left\{ g : \|g\|_{\mathcal{B}_R^p(\Omega)} < \infty \right\}.$$

It is worth making some comments on the definition above. Our definition of Barron space adapts a similar definition in [8] (see also [1]) with several important modifications for the purpose of PDE analysis. First we require that the $w$-marginal of the probability measure $\rho$ has compact support in order to control the derivatives of a Barron function defined in (2.2); in fact differentiating the integral of (2.2) leads to an integral of the product of the ridge function with $w$ (or its powers) and enforcing $\rho$ has a compact $w$-marginal thus controls the Barron norm of the derivatives of $u_\rho$. In addition, our definition of Barron norm only involves the $p$-th moment of $\rho$ with respect to $a$ parameter whereas the Barron norm in [8] takes the moments in all parameters into account. This is because [8] uses the unbounded ReLU activation function, which requires the moment condition in all parameters to make the integral in (2.2) well-defined; whereas we will only consider bounded $\sigma$ (see Assumption 2.3) and the integral is guaranteed to be finite under such assumption.

Both our notion of Barron space and the one in [8] are motivated by the seminal work of Barron [2] where he proved that if the Fourier transform $\mathcal{F}(f)$ of a function $f$ satisfies that

$$\int_{\mathbb{R}^d} |\mathcal{F}(f)(\xi)||\xi|d\xi < \infty,$$

then there exists a two-layer network $u_k$ with $k$ hidden neurons such that $\|f - u_k\|_{L^2(\Omega)} \leq Ck^{-\frac{1}{2}}$. Since Barron's original function class is defined via the Fourier transform, we call such function class the *spectral Barron space* to distinguish it from our Barron space based on the probability integral. We refer to [24, 3, 39, 40, 28] for recent developments on the spectral Barron space.

As we investigate the solution theory of the second-order PDE in the Barron space, we expect to differentiate the integral representation (2.2) up to the second order. Therefore, we assume that the activation function $\sigma$ as well as its first-order and second-order derivatives are all bounded in $\mathbb{R}$.

**Assumption 2.3.** $\sigma : \mathbb{R} \to \mathbb{R}$ *is smooth with* $C_0 := \sup_{y \in \mathbb{R}} |\sigma(y)| < \infty$, $C_1 := \sup_{y \in \mathbb{R}} |\sigma'(y)| < \infty$, *and* $\sup_{y \in \mathbb{R}} |\sigma''(y)| < \infty$.

Thanks to the Hölder inequality, it is clear that $\mathcal{B}_R^p(\Omega) \subset \mathcal{B}_R^q(\Omega)$ when $p \leq q$. The following useful proposition (see also [8, Proposition 1]) shows that the reverse is also true and that the Barron norms and the Barron spaces are in fact independent of $p$.

**Proposition 2.4.** *For any function* $g \in \mathcal{B}_R^1(\Omega)$, *it holds that* $\|g\|_{\mathcal{B}_R^\infty(\Omega)} = \|g\|_{\mathcal{B}_R^p(\Omega)} = \|g\|_{\mathcal{B}_R^1(\Omega)}$ *for any* $1 \leq p \leq \infty$. *As a consequence,* $\mathcal{B}_R^\infty(\Omega) = \mathcal{B}_R^p(\Omega) = \mathcal{B}_R^1(\Omega)$ *for* $1 \leq p \leq \infty$.

The proof of Proposition 2.4 can be found in Appendix B.

The most important property that makes Barron functions distinct from Sobolev or Hölder functions is that they can be approximated by two-layer neural networks with a dimension-independent approximation rate in $H^1$ norm as shown in Theorem 2.5.

**Theorem 2.5** (Approximation theorem in $H^1$ norm)**.** *Suppose that Assumption 2.3 holds and that* $g \in \mathcal{B}_R^1(\Omega)$. *Then for any open bounded subset* $\Omega_0 \subset \Omega$ *and any* $k \in \mathbb{N}_+$, *there exists* $\{(a_i, w_i, b_i)\}_{i=1}^k$ *satisfying*

$$\left\| \frac{1}{k} \sum_{i=1}^k a_i \sigma(w_i^\top x + b_i) - g(x) \right\|_{H^1(\Omega_0)}^2 \leq \frac{2(C_0^2 + R^2 C_1^2) m(\Omega_0) \|g\|_{\mathcal{B}_R^1(\Omega)}^2}{k}, \qquad (2.3)$$

*where* $C_0$ *and* $C_1$ *are the constants in Assumption 2.3, and* $m(\Omega_0)$ *is the Lebesgue measure of* $\Omega_0$.

Theorem 2.5 provides an $H^1$-approximation rate for Barron functions defined by the integral representation (2.2). The proof is deferred to Appendix B. Similar approximation results in the sense of $L^2$ for Barron functions (including formulations based on spectrum and integral representation) have been proved in [2, 24, 3, 39, 8]. $H^1$-approximation results for spectral Barron functions were previously obtained in [40] and [28].

## 2.3 Main theorems

To state our main theorems, we need to make some additional complexity assumption on the coefficients $A, c$ and the source term $f$ of the PDE (1.1), which is reasonable as otherwise there is no hope that the solution would lie in a smaller function class.

**Assumption 2.6.** *For some $R_A, R_c, R_f \in (0, +\infty)$, we have $\ell_A := \max_{1 \leq i,j \leq d} \|A_{ij}\|_{\mathcal{B}^1_{R_A}(\mathbb{R}^d)} < \infty$, $\ell_c := \|c\|_{\mathcal{B}^1_{R_c}(\mathbb{R}^d)} < \infty$, and $\ell_f := \|f\|_{\mathcal{B}^1_{R_f}(\mathbb{R}^d)} < \infty$.*

We remark that Assumption 2.6 is compatible with our earlier Assumption 2.1 on the coefficients $A, c$ and the source $f$. In fact, it is easy to see that constant coefficients $A, c$ satisfy both assumptions if $\text{im}(\sigma) \neq \{0\}$, i.e., $\sigma$ is not constantly zero. As for $f$, we provide in Proposition A.1 of Appendix A a concrete class of $f$ that satisfies both assumptions.

We also need two additional technical assumptions on the activation function.

**Assumption 2.7.** *The function $h : \mathbb{R}^2 \to \mathbb{R}$, $(y_1, y_2) \mapsto \sigma(y_1)\sigma(y_2)$ satisfies that $\ell_m := \|h\|_{\mathcal{B}^1_{R_m}(\mathbb{R}^2)} < \infty$, for some $R_m \in (0, +\infty)$.*

**Assumption 2.8.** *It holds that $\ell_{d,1} := \|\sigma'\|_{\mathcal{B}^1_{R_{d,1}}(\mathbb{R})} < \infty$ and $\ell_{d,2} := \|\sigma''\|_{\mathcal{B}^1_{R_{d,2}}(\mathbb{R})} < \infty$, for some $R_{d,1}, R_{d,2} \in (0, +\infty)$.*

Assumption 2.7 and Assumption 2.8 guarantee that Barron spaces are closed under multiplication and differentiations (up to the second order) respectively; see Lemma 3.3 (iii)-(iv) for a precise statement. These operations and the associated closeness will be useful for constructing approximation to the exact solution $u^*$ of the PDE (1.1) in Barron spaces. Proposition A.2 shows that Assumption 2.7 and Assumption 2.8 hold for a relatively large class of activation functions including cosine.

With the preparations above, we are ready to state our main theorems below. The first main theorem concerns the complexity estimate of the exact solution $u^*$ in the Barron space.

**Theorem 2.9.** *Suppose that Assumption 2.1, 2.3, 2.6, 2.7, and 2.8 hold. For any $\epsilon \in (0, 1/2)$, there exists $u \in \mathcal{B}^1_R(\mathbb{R}^d)$ with $R \leq \gamma_1 \left(\frac{1}{\epsilon}\right)^{\gamma_2}$ and $\|u\|_{\mathcal{B}^1_R(\mathbb{R}^d)} \leq \beta_1 \left(\frac{d}{\epsilon}\right)^{\beta_2|\ln \epsilon|}$, such that $\|u - u^*\|_{H^1(\mathbb{R}^d)} \leq \epsilon$. Here $\gamma_1, \gamma_2, \beta_1$, and $\beta_2$ only depend on $\|f\|_{H^{-1}(\mathbb{R}^d)}$ and constants in Assumptions 2.1, 2.6, 2.7, and 2.8.*

*Furthermore, if $\sigma = \cos$, then $\|u - u^*\|_{H^1(\mathbb{R}^d)} \leq \epsilon$ can be achieved with $R \leq \gamma_1'|\ln \epsilon|$ and $\|u\|_{\mathcal{B}^1_R(\mathbb{R}^d)} \leq \beta_1' d^{\beta_2'|\ln \epsilon|}$, where $\gamma_1', \beta_1'$, and $\beta_2'$ only depend on $\|f\|_{H^{-1}(\mathbb{R}^d)}$ and constants in Assumption 2.1 and 2.6.*

Theorem 2.9 shows that the exact solution $u^*$ is $\epsilon$-close (in the sense of $H^1$) to a Barron function $u \in \mathcal{B}^1_R(\mathbb{R}^d)$. In addition, the Barron norm of $u$ grows at most polynomially in $d$, indicating that the complexity of $u$ dose not suffer from the CoD. Also the complexity estimate gets substantially improved when the activation function is cosine. In fact, advantages of periodic activation functions have been empirically observed in some earlier works, see e.g., [42]. It remains an open question whether results similar to Theorem 2.9 can be established for activation functions that do not satisfy Assumption 2.7 and Assumption 2.8. This will be investigated in future works.

Thanks to Theorem 2.5 and Theorem 2.9, it is easy to conclude that the PDE solution $u^*$ can be approximated on any bounded subset $\Omega \subset \mathbb{R}^d$ using two-layer neural networks with the number of hidden neurons $k$ scaling at most polynomially in $d$.

**Theorem 2.10.** *Under the same assumptions as in Theorem 2.9, given any $\epsilon \in (0, 1/2)$ and any open bounded subset $\Omega \subset \mathbb{R}^d$, there exists a two-layer neural network $u_k(x)$ with $k \leq \gamma m(\Omega) \left(\frac{d}{\epsilon}\right)^{\beta|\ln \epsilon|}$ such that $\|u_k - u^*\|_{H^1(\Omega)} \leq \epsilon$, where $\gamma$ and $\beta$ only depend on $\|f\|_{H^{-1}(\mathbb{R}^d)}$ and constants in Assumptions 2.1, 2.3, 2.6, 2.7, and 2.8.*

*Furthermore, if $\sigma = \cos$, then $\|u_k - u^*\|_{H^1(\Omega)} \leq \epsilon$ can be achieved with $k \leq \gamma' m(\Omega) d^{\beta'|\ln \epsilon|}$, where $\gamma'$ and $\beta'$ only depend on $\|f\|_{H^{-1}(\mathbb{R}^d)}$ and constants in Assumptions 2.1, 2.3, and 2.6.*

## 3 Proofs of the main results

We sketch the proof ideas in this section and present the full details in the Appendix.

## 3.1 Preconditioned functional iterative scheme

The key ingredient of our proof of Theorem 2.9 is a functional iterative scheme for solving the elliptic PDE, which can be viewed as an infinite dimensional analog of the preconditioned steepest descent algorithm to solve linear algebra equations. Recall when solving the linear equation $Ax = b$ with $A \in \mathbb{R}^{n \times n}$ and $x, b \in \mathbb{R}^n$, the preconditioned steepest descent algorithm [13] runs the iteration

$$x_{t+1} = x_t - \alpha P(Ax_t - b),$$

where $P$ is a preconditioning matrix, $\alpha$ is the step size, and $t = 0, 1, 2, \cdots$ indicates the iteration index. The purpose of the preconditioned iteration is to reduce the condition number of the iteration $\kappa(PA)$ by choosing a suitable $P$ and hence accelerate the convergence of the iterative algorithm.

In the case of solving the elliptic PDE (1.1), we generalize the preconditioned steepest descent iteration to the functional setting by considering the following iteration scheme in $H^1(\mathbb{R}^d)$:

$$u_{t+1} = u_t - \alpha(I - \Delta)^{-1}(\mathcal{L}u_t - f), \tag{3.1}$$

where the inverse operator $(I - \Delta)^{-1}$ plays the role of preconditioner. As a matter of fact, we will show that the condition number of $(I - \Delta)^{-1}\mathcal{L}$ is bounded and this directly implies that the iterative scheme (3.1) converges exponentially to the exact solution $u^*$. Indeed, we have the following contraction estimate for the iteration (3.1), whose proof can be found in Appendix C.

**Proposition 3.1.** *Recall the constants $a_{\min}, a_{\max}, c_{\min}, c_{\max}$ defined in Assumption 2.1. For any $\alpha > 0$ and any $u \in H^1(\mathbb{R}^d)$,*

$$\left\|(I - \alpha(I - \Delta)^{-1}\mathcal{L})u\right\|_{H^1(\mathbb{R}^d)} \leq \Lambda(\alpha)\|u\|_{H^1(\mathbb{R}^d)}, \tag{3.2}$$

*where the contraction factor $\Lambda(\alpha) = \sup_{\lambda \in [\lambda_{\min}, \lambda_{\max}]} |1 - \alpha\lambda|$ with $\lambda_{\min} = \min\{a_{\min}, c_{\min}\}$ and $\lambda_{\max} = \max\{a_{\max}, c_{\max}\}$.*

In particular, minimizing $\Lambda(\alpha)$ with respect to the step size $\alpha$ yields an optimal choice of step size

$$\alpha_* := \frac{2}{\lambda_{\min} + \lambda_{\max}}.$$

With $\alpha = \alpha_*$ in (3.2), we obtain that

$$\left\|\left(I - \frac{2}{\lambda_{\min} + \lambda_{\max}}(I - \Delta)^{-1}\mathcal{L}\right)u\right\|_{H^1(\mathbb{R}^d)} \leq \frac{\lambda_{\max} - \lambda_{\min}}{\lambda_{\max} + \lambda_{\min}}\|u\|_{H^1(\mathbb{R}^d)}. \tag{3.3}$$

As a direct consequence, we obtain the following estimate for the number of iterations required to achieve a given error tolerance.

**Corollary 3.2.** *Let $u^*$ be the exact solution of the PDE* (1.1). *Under Assumption 2.1, consider the iteration scheme* (3.1) *with $\alpha = \alpha_* = \frac{2}{\lambda_{\min} + \lambda_{\max}}$. Then for any*

$$T \geq \left(\ln \frac{\lambda_{\max} + \lambda_{\min}}{\lambda_{\max} - \lambda_{\min}}\right)^{-1} \ln \frac{\|u_0 - u^*\|_{H^1(\mathbb{R}^d)}}{\epsilon},$$

*the iterate $u_T$ satisfies $\|u_T - u^*\|_{H^1(\mathbb{R}^n)} \leq \epsilon$.*

Let us remark that the idea of using iterative scheme to establish neural network representation results of solutions to PDEs is not new, see e.g., [23, 30], similar ideas have been also used to construct neural network architectures inspired from iterative schemes, see e.g., [43, 12]. Closely related to our setting, the work [30] uses a steepest descent iteration with the right hand side of the equation assumed to be in the span of first several eigenfunctions of the elliptic operator, while [23] considered general right hand side, but only after discretization which also effectively truncates the problem onto a finite dimensional subspace. These restrictions were made to limit the condition number of the iteration. Unlike those works using standard steepest descent iterations, by using the preconditioning technique, we can deal with general right hand side without restricting to a finite-dimensional subspace.

## 3.2 Algebra of Barron functions and representation of the solution

Corollary 3.2 in the previous subsection shows that we can obtain an approximate solution by running the iteration (3.1). To complete the proof of Theorem 2.9, we show in this subsection that the iteration (3.1) can be carried out in the Barron space $\mathcal{B}_R^1(\mathbb{R}^d)$, i.e. each iteration $u_t \in \mathcal{B}_R^1(\mathbb{R}^d)$ (with the support radius $R$ potentially depending on $t$). To this end, we first need to establish the closeness of Barron space under function operations involved in the iteration. In fact, by decomposing each of the iteration step in (3.1) into two steps, we can write

$$\begin{cases} v_t = \mathcal{L}u_t - f = -\sum_{i,j}(\partial_i A_{ij}\partial_j u_t + A_{ij}\partial_{ij}u_t) + cu_t - f, \\ u_{t+1} = u_t - \alpha(I - \Delta)^{-1}v_t. \end{cases} \tag{3.4}$$

Thus, to show that the iterate $u_t$ remains in Barron space, it suffices to establish that addition, scalar multiplication, product, differentiation, and action of $(I - \Delta)^{-1}$ are closed in the Barron space. The closedness of Barron functions under those operations are not only useful for proving our main results, but also of its own interest. The next two lemmas summarize the algebras and the stability estimate of the inverse $(I - \Delta)^{-1}$ in the Barron space. Their proofs can be found in Appendix D.

**Lemma 3.3** (Algebras in Barron spaces). *The followings hold:*

(i) *(Addition) Suppose that $\|g_i\|_{\mathcal{B}_{R_i}^1(\mathbb{R}^d)} < \infty$, $i = 1, 2, \ldots, k$. Then $\|g_1 + \cdots + g_k\|_{\mathcal{B}_R^1(\mathbb{R}^d)} \leq \sum_{1 \leq i \leq k} \|g_i\|_{\mathcal{B}_{R_i}^1(\mathbb{R}^d)}$, where $R = \max_{1 \leq i \leq k} R_i$.*

(ii) *(Scalar multiplication) Suppose that $\|g\|_{\mathcal{B}_R^1(\mathbb{R}^d)} < \infty$ and that $\lambda \in \mathbb{R}$. Then $\|\lambda g\|_{\mathcal{B}_R^1(\mathbb{R}^d)} = |\lambda| \|g\|_{\mathcal{B}_R^1(\mathbb{R}^d)}$.*

(iii) *(Product) Suppose that Assumption 2.3 and Assumption 2.7 hold and that $\|g_i\|_{\mathcal{B}_{R_i}^1(\mathbb{R}^d)} < \infty$ for $i = 1, 2$. Then $\|g_1 g_2\|_{\mathcal{B}_R^1(\mathbb{R}^d)} \leq \ell_m \|g\|_{\mathcal{B}_{R_1}^1(\mathbb{R}^d)} \|g\|_{\mathcal{B}_{R_2}^1(\mathbb{R}^d)}$, where $R = R_m(R_1 + R_2)$ with $R_m$ and $\ell_m$ being constants in Assumption 2.7.*

(iv) *(Derivatives) Suppose that Assumption 2.3 and Assumption 2.8 hold and that $\|g\|_{\mathcal{B}_R^1(\mathbb{R}^d)} < \infty$ with $R < \infty$. Then $\|\partial_i g\|_{\mathcal{B}_{R_{d,1}R}^1(\mathbb{R}^d)} \leq \ell_{d,1}R\|g\|_{\mathcal{B}_R^1(\mathbb{R}^d)}$ and $\|\partial_{ij}g\|_{\mathcal{B}_{R_{d,2}R}^1(\mathbb{R}^d)} \leq \ell_{d,2}R^2\|g\|_{\mathcal{B}_R^1(\mathbb{R}^d)}$ for any $i, j \in \{1, 2, \ldots, d\}$, where $R_{d,1}, R_{d,2}, \ell_{d,1}$, and $\ell_{d,2}$ are constants in Assumption 2.8.*

**Lemma 3.4** (Applying $(I - \Delta)^{-1}$ on Barron functions). *Suppose that $\|g\|_{\mathcal{B}_R^1(\mathbb{R}^d)} < \infty$. Then $\left\|(I - \Delta)^{-1}g\right\|_{\mathcal{B}_R^1(\mathbb{R}^d)} \leq \|g\|_{\mathcal{B}_R^1(\mathbb{R}^d)}$.*

we include a proof for Lemma 3.4 in Appendix D that uses similar arguments from [9], though the analysis in [9] is for $d \geq 3$. The lemmas above lead to the following recursive estimate on the Barron norm of $u_t$.

**Lemma 3.5.** *Suppose that Assumption 2.3, Assumption 2.7, and Assumption 2.8 hold. If $\|u\|_{\mathcal{B}_{R_{u,t}}^1} < \infty$ with $R_{u,t} < \infty$, then $u_{t+1}$ defined in (3.1) or (3.4) satisfies that*

$$\|u_{t+1}\|_{\mathcal{B}_{R_{u,t+1}}^1(\mathbb{R}^d)} \leq \left(\alpha\ell_m\ell_A(\ell_{d,1}^2 R_A R_{u,t} + \ell_{d,2}R_{u,t}^2)d^2 + \alpha\ell_m\ell_c + 1\right)\|u_t\|_{\mathcal{B}_{R_{u,t}}^1(\mathbb{R}^d)} + \alpha\ell_f, \tag{3.5}$$

*for any*

$$R_{u,t+1} \geq \max\{R_m R_{d,1}(R_{u,t} + R_A), R_m(R_{d,2}R_{u,t} + R_A), R_m(R_{u,t} + R_c), R_{u,t}, R_f\}. \tag{3.6}$$

The proof of Lemma 3.5 is deferred to Appendix D. One observation is that the amplification factor of the Barron norm in Lemma 3.5 increases as the support radius $R$ increases. The reason is that differentiating the function would introduce components of $w$ and hence the amplification depends on how large $\|w\|$ can be and thus the support of the measure.

One possible direction to improve the estimate is to realize that the preconditioner $(I - \Delta)^{-1}$ can counteract the action of taking derivatives. It is indeed possible to to remove the $R$ dependence from the amplification factor, at least for some specific activation functions, through a more careful analysis. In particular, we have the following lemma for the cosine activation function, the proof of which can also be found in Appendix D.

**Lemma 3.6.** *Suppose that Assumption 2.6 holds. If $\sigma = \cos$ and $\|u\|_{\mathcal{B}^1_{R_{u,t}}(\mathbb{R}^d)} < \infty$ with $R_{u,t} < \infty$, then $u_{t+1}$ defined in (3.1) or (3.4) satisfies*

$$\|u_{t+1}\|_{\mathcal{B}^1_{R_{t+1}}(\mathbb{R}^d)} \leq \left(6\alpha\ell_A \max\{R_A^2, 1\}d^2 + \alpha\ell_c + 1\right) \|u_t\|_{\mathcal{B}^1_{R_{u,t}}(\mathbb{R}^d)} + \alpha\ell_f, \qquad (3.7)$$

*for any*

$$R_{u,t+1} \geq R_{u,t} + \max\{R_A, R_c, R_f\}. \qquad (3.8)$$

Lemma 3.5 and Lemma 3.6 estimate the amplification of the Barron norm in each iteration of (3.1). Combining them with the control of number of iterations, Corollary 3.2, we are ready to finish the proof of Theorem 2.9.

*Proof of Theorem 2.9.* Fix $u_0 = 0$ and $\alpha = \frac{2}{\lambda_{\min} + \lambda_{\max}}$. According to Corollary 3.2, it holds that $\|u_T - u^*\|_{H^1(\mathbb{R}^n)} \leq \epsilon$ for any

$$T \geq \left(\ln \frac{\lambda_{\max} + \lambda_{\min}}{\lambda_{\max} - \lambda_{\min}}\right)^{-1} \ln \frac{\|u^*\|_{H^1(\mathbb{R}^n)}}{\epsilon}.$$

Moreover, thanks to the estimate

$$\lambda_{\min} \|u^*\|_{H^1(\mathbb{R}^d)}^2 \leq \int A\nabla u^* \cdot \nabla u^* dx + \int c|u^*|^2 dx = \int f u^* dx \leq \|f\|_{H^{-1}(\mathbb{R}^d)} \|u^*\|_{H^1(\mathbb{R}^d)},$$

we have $\|u^*\|_{H^1(\mathbb{R}^d)} \leq \frac{1}{\lambda_{\min}} \|f\|_{H^{-1}(\mathbb{R}^d)}$. Therefore, it suffices to take

$$T = \left\lceil \left(\ln \frac{\lambda_{\max} + \lambda_{\min}}{\lambda_{\max} - \lambda_{\min}}\right)^{-1} \ln \frac{1}{\epsilon} + \left(\ln \frac{\lambda_{\max} + \lambda_{\min}}{\lambda_{\max} - \lambda_{\min}}\right)^{-1} \ln \frac{\|f\|_{H^{-1}(\mathbb{R}^d)}}{\lambda_{\min}} \right\rceil.$$

Set $R_{u,0} = \max\{R_A, R_c, R_f, 1\}$ and $R_{u,t+1} = \max\{2R_m R_{d,1}, 2R_m R_{d,2}, 2R_m, 1\} \cdot R_{u,t} \geq R_{u,t}$. Then (3.6) is satisfied for any $t$. Let us define a sequence $\{X_t\}_{t\geq 0}$ via $X_0 = 1$ and $X_{t+1} = \left(\alpha\ell_m \ell_A(\ell_{d,1}^2 + \ell_{d,2}) + \frac{\alpha(\ell_m \ell_c + \ell_f) + 1}{d^2}\right) R_{u,t}^2 d^2 \cdot X_t$. By (3.5), we have $\|u_t\|_{\mathcal{B}^1_{R_{u,t}}(\mathbb{R}^d)} \leq X_t$ for any $t$. Therefore, it holds that

$$R_{u,T} = \max\{R_A, R_c, R_f, 1\} \cdot \max\{2R_m R_{d,1}, 2R_m R_{d,2}, 2R_m, 1\}^T,$$

and that

$$\|u_T\|_{\mathcal{B}^1_{R_{u,T}}(\mathbb{R}^d)} \leq X_T$$

$$= \left(\alpha\ell_m \ell_A(\ell_{d,1}^2 + \ell_{d,2}) + \frac{\alpha(\ell_m \ell_c + \ell_f) + 1}{d^2}\right)^T d^{2T} (R_{u,0} \cdots R_{u,T-1})^2$$

$$\leq \left(\alpha\ell_m \ell_A(\ell_{d,1}^2 + \ell_{d,2}) + \frac{\alpha(\ell_m \ell_c + \ell_f) + 1}{d^2}\right)^T d^{2T}$$

$$\cdot (\max\{R_A, R_c, R_f, 1\})^T \cdot \max\{2R_m R_{d,1}, 2R_m R_{d,2}, 2R_m, 1\}^{T^2}.$$

The first part of Theorem 2.9 is established by setting $u = u_T$ and $R = R_{u,T}$.

If $\sigma = \cos$, (3.8) is satisfied by setting

$$R_{u,t} = \max\{R_A, R_c, R_f\} \cdot t.$$

Define $Y_0 = 0$ and $Y_{t+1} = \left(6\alpha\ell_A \max\{R_A^2, 1\}d^2 + \alpha\ell_c + 1\right) Y_t + \alpha\ell_f$. By (3.7), we obtain that $\|u_t\|_{\mathcal{B}^1_{R_{u,t}}(\mathbb{R}^d)} \leq Y_t$ for any $t$, and in particular that

$$\|u_T\|_{\mathcal{B}^1_{R_{u,T}}(\mathbb{R}^d)} \leq Y_T = \frac{\alpha\ell_f \left(\left(6\alpha\ell_A \max\{R_A^2, 1\}d^2 + \alpha\ell_c + 1\right)^T - 1\right)}{6\alpha\ell_A \max\{R_A^2, 1\}d^2 + \alpha\ell_c},$$

which finishes the proof by setting $u = u_T$ and $R = R_{u,T}$. $\qquad\square$

Theorem 2.10 is then a corollary of Theorem 2.9 and Theorem 2.5 (the approximation theorem).

*Proof of Theorem 2.10.* Theorem 2.10 follows directly from applying Theorem 2.5 with error tolerance $\epsilon/2$ and applying Theorem 2.9 with error tolerance $\epsilon/2$. $\qquad\square$

# 4    Conclusion

In this work, we establish the approximation rate for the solution of a second-order elliptic PDE by a Barron function and by a two-layer neural network. Under the assumption that the coefficients and the source of the PDE are all in the Barron spaces with some compact support property on the underlying probability measure, the approximation rate is shown to depend at most polynomially on the dimension. Therefore, our results indicate that even a neural network as simple as a two-layer network with a single activation function can have adequate representation ability to encode the solution of an elliptic PDE, without incurring the CoD. Our result provides theoretical guarantee for numerical methods for solving high-dimensional PDEs using neural networks.

For future directions, it is of interest to extend the functional analysis framework to more general activation functions (such as unbounded ones) and more general neural network architectures. One interesting direction is to establish depth separation result for representing PDE solutions. Our analysis also indicates some potential benefit of using periodic activation function such as cosine in terms of approximation, further studies and understanding of the choice of activation function and architecture are crucial. Moreover, while we focus on approximation error, generalization error and analysis of training should also be considered in future works.

It is possible to extend the approximation results to a wider range of high-dimensional PDEs such as parabolic PDEs, PDE eigenvalue problems, and nonlinear equations such as those arise from control theory. The analysis tools and characterization of Barron space we establish in this work would be useful for these future studies.

## Acknowledgments and Disclosure of Funding

The work of Z.C. and J.L. is supported in part by the National Science Foundation via grants DMS-2012286 and CCF-1934964. Y.L. thanks the National Science foundation for its support through the award DMS-2107934.

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
