## A    Validity of assumptions

In this section, we show that the set of right hand side $f$ satisfying Assumption 2.1 and Assumption 2.6, and the set of activation functions $\sigma$ satisfying Assumption 2.7 and Assumption 2.8 are not empty. We first give a concrete example of $f$ that fulfills Assumption 2.1 and Assumption 2.6.

**Proposition A.1.** *Suppose that $\sigma = \cos$. Consider a complex-valued function $f_0 \in L^2(\mathbb{R}^d) \cap L^1(\mathbb{R}^d)$ that is compactly supported, it holds that $f = Re(\mathcal{F}^{-1}f_0) \in L^2(\mathbb{R}^d) \cap \mathcal{B}^1_{R_f}(\mathbb{R}^d)$ for some $0 < R_f < +\infty$, where $\mathcal{F}^{-1}$ is the inverse Fourier transform.*

We first introduce some concepts and facts, which will be useful for proving Proposition A.1 and other results.

**Push-forward measure**    Consider two measure spaces $(X, \mathcal{F}_X)$ and $(Y, \mathcal{F}_Y)$ and a measurable map $T : X \to Y$. Given any measure $\mu_X$ on $(X, \mathcal{F}_X)$, one can define the push-forward measure $\mu_Y = T_*\mu_X$ via

$$\mu_Y(A) = \mu_X(T^{-1}A), \quad A \in \mathcal{F}_Y.$$

If $\mu_X$ is a probability measure, then $\mu_Y = T_*\mu_X$ is also a probability measure. For any integrable function $g : Y \to \mathbb{R}$, it holds that

$$\int_Y g(y)\mu_Y(dy) = \int_X g(T(x))\mu_X(dx).$$

**Fourier transform**    Let $\hat{f} := \mathcal{F}(f)(\xi)$ be the Fourier transform of $f \in L^1(\mathbb{R}^d)$, i.e.

$$\hat{f}(\xi) = (2\pi)^{-\frac{d}{2}} \int_{\mathbb{R}^d} e^{-ix\cdot\xi} f(x)dx.$$

Denote by $\mathcal{F}^{-1}(\hat{f})$ the inverse Fourier transform of $\hat{f}$, given by

$$f(x) = \mathcal{F}^{-1}(\hat{f})(x) := (2\pi)^{-\frac{d}{2}} \int_{\mathbb{R}^d} e^{ix\cdot\xi} \hat{f}(\xi)d\xi,$$

if $\hat{f} \in L^1(\mathbb{R}^d)$. The Fourier transform and the inverse Fourier transform can be extended to the space of tempered distributions, i.e., the dual space $\mathcal{S}'_d$ of the Schwartz space $\mathcal{S}_d$. Recall the Parseval's identity for $f \in L^2(\mathbb{R}^d)$: $\|f\|^2_{L^2(\mathbb{R}^d)} = \|\hat{f}\|^2_{L^2(\mathbb{R}^d)}$.

*Proof of Proposition A.1.* We use techniques from [2] to prove this proposition. Let $f_0(\xi) = e^{i\theta(\xi)}F(\xi)$, where $F(\xi) = |f_0(\xi)|$. It holds that

$$\begin{aligned}
f(x) &= \text{Re}\left((\mathcal{F}^{-1}f_0)(x)\right) \\
&= \text{Re}\left((2\pi)^{-\frac{d}{2}} \int e^{i\xi^\top x} f_0(\xi)d\xi\right) \\
&= \text{Re}\left((2\pi)^{-\frac{d}{2}} \int e^{i(\xi^\top x + \theta(\xi))} F(\xi)d\xi\right) \\
&= (2\pi)^{-\frac{d}{2}} \int \cos(\xi^\top x + \theta(\xi))\mu(d\xi) \\
&= \int \frac{a_0}{(2\pi)^{\frac{d}{2}}} \cos(\xi^\top x + \theta(\xi))\mu'(d\xi), \quad x \in \mathbb{R}^d,
\end{aligned}$$

where the measure $\mu$ is defined via $\mu(A) = \int_A F(\xi)d\xi$, $a_0 = \mu(\mathbb{R}^d) = \int_{\mathbb{R}^d} F(\xi)d\xi$, and $\mu' = \mu/a_0$ is a probability measure. Note that $a_0 < \infty$ because $f_0 \in L^1(\mathbb{R}^d)$. Set the push-forward measure $\rho = T_*\mu'$ where $T : \mathbb{R}^d \to \mathbb{R} \times \mathbb{R}^d \times \mathbb{R}$, $\xi \mapsto \left(\frac{a_0}{(2\pi)^{\frac{d}{2}}}, \xi, \theta(\xi)\right)$. Then we obtain that

$$f(x) = \int a \cos(w^\top x + b)\rho(da, dw, db), \quad x \in \mathbb{R}^d.$$

Note that $f_0$ is compactly supported. There exists some $R_f < \infty$ such that $f_0$ is supported on $\overline{B}_{R_f}^d$. By definition, we know that $\mu$ and $\mu'$ are both supported on $\overline{B}_{R_f}^d$, and hence that $\rho$ is supported on $\mathbb{R} \times \overline{B}_{R_f}^d \times \mathbb{R}$. Therefore, we obtain that

$$\|f\|_{\mathcal{B}_{R_f}^1(\mathbb{R}^d)} \leq \int |a| \rho(da, dw, db) = \frac{a_0}{(2\pi)^{\frac{d}{2}}} < \infty,$$

which implies that $f \in \mathcal{B}_{R_f}^1(\mathbb{R}^d)$. Moreover, it follows from $\|f\|_{L^2(\mathbb{R}^d)} \leq \|\mathcal{F}^{-1} f_0\|_{L^2(\mathbb{R}^d)} = \|f_0\|_{L^2(\mathbb{R}^d)} < \infty$ that $f \in L^2(\mathbb{R}^d)$. $\qquad\square$

The next proposition shows that functions who are band-limited, smooth, and periodic satisfy Assumption 2.7 and Assumption 2.8.

**Proposition A.2.** *If $\sigma$ is band-limited, smooth, and periodic, then Assumption 2.7 and Assumption 2.8 are satisfied. In particular, $\sigma = \cos$ satisfies Assumption 2.7 and Assumption 2.8.*

*Proof.* The smoothness and periodicity of $\sigma$ imply that $\sigma$ is equal to its Fourier series (recall that the smoothness of $\sigma$ leads to fast decay of the Fourier coefficients and hence the uniform convergence of Fourier series). Since $\sigma$ is band-limited, its Fourier series only have finitely many terms as $\mathcal{F}(e^{iax}) = \delta(\xi - a)$. Therefore, there exists $k \in \mathbb{N}_+$ and $\{(a_i, w_i, b_i)\}_{i=1}^k \subset (\mathbb{R}\backslash\{0\}) \times \mathbb{R} \times \mathbb{R}$ such that $\sigma(y) = \sum_{i=1}^k a_i \cos(w_i y + b_i)$. Without loss of generality, let us assume that $|w_1| < |w_2| < \cdots < |w_k|$. We also assume that $\sigma$ is not a constant as otherwise the results are trivial. Notice that $\cos'(y) = \cos(y + \pi/2)$, $\cos''(y) = \cos(y + \pi)$, and $\cos(y_1)\cos(y_2) = \frac{1}{2}\cos(y_1 + y_2) + \frac{1}{2}\cos(y_1 - y_2)$. Therefore, it suffices to show that there exists $m \in \mathbb{N}_+$ and $\{(\gamma_i, \xi_i, \eta_i)\}_{i=1}^m \subset \mathbb{R} \times \mathbb{R} \times \mathbb{R}$, such that $\sum_{i=1}^m \gamma_i \sigma(\xi_i y + \eta_i) = \cos(y)$. This is trivial if $k = 1$.

Now we consider $k \geq 2$. If $|w_1| \neq 0$, then it holds that

$$\sigma\left(y + \frac{2\pi}{|w_k|}\right) - \sigma(y) = \sum_{i=1}^{k-1}\left(a_i \cos\left(w_i y + \frac{2\pi w_i}{|w_k|} + b_i\right) - a_i \cos(w_i y + b_i)\right)$$
$$= \sum_{i=1}^{k-1} a_i' \cos(w_i y + b_i'),$$

where $a_i' \neq 0$ for $1 \leq i \leq k - 1$. If $w_1 = 0$, then let us choose $y_0 \notin \cup_{2 \leq i \leq k}\{2\ell\pi/w_i : \ell \in \mathbb{Z}\}$, then it holds that

$$\sigma(y + y_0) - \sigma(y) = \sum_{i=2}^k (a_i \cos(w_i y + w_i y_0 + b_i) - a_i \cos(w_i y + b_i))$$
$$= \sum_{i=2}^k a_i' \cos(w_i y + b_i'),$$

where $a_i' \neq 0$ for $2 \leq i \leq k$. Both cases are reduced to $k - 1$. Then we can finish the proof by induction. $\qquad\square$

# B  Proofs for Section 2.2

In this section, we present proofs of some properties of Barron norms and Barron spaces, say Proposition 2.4 and Theorem 2.5. Note that the proof techniques are not new. They are borrowed from [8] and [2].

*Proof of Proposition 2.4.* This proof is modified from [8], especially the proof in [8, Section 2.5.1]. If $\|g\|_{\mathcal{B}_R^1(\Omega)} = \infty$, it is clear that $\|g\|_{\mathcal{B}_R^p(\Omega)} = \infty$ for any $1 \leq p \leq \infty$. Thus, we assume that $\|g\|_{\mathcal{B}_R^1(\Omega)} < \infty$. By Hölder's inequality, it holds that

$$\|g\|_{\mathcal{B}_R^1(\Omega)} \leq \|g\|_{\mathcal{B}_R^p(\Omega)} \leq \|g\|_{\mathcal{B}_R^\infty(\Omega)}.$$

Therefore, it suffices to show that $\|g\|_{\mathcal{B}_R^\infty(\Omega)} \leq \|g\|_{\mathcal{B}_R^1(\Omega)}$. Consider any $\epsilon > 0$, there exists a probability measure $\rho$ supported on $\mathbb{R} \times \overline{B}_R^d \times \mathbb{R}$ such that

$$g(x) = \int a\sigma(w^\top x + b)\rho(da, dw, db), \quad \forall x \in \Omega,$$

and that

$$\ell := \int |a|\rho(da, dw, db) \leq \|g\|_{\mathcal{B}_R^1(\Omega)} + \epsilon.$$

Define a new probability measure $\mu$ supported on $\{\ell, -\ell\} \times \overline{B}_R^d \times \mathbb{R}$ via

$$\mu(\{\ell\} \times A) = \frac{1}{\ell} \int_{(0,+\infty) \times A} |a|\rho(da, dw, db),$$

and

$$\mu(\{-\ell\} \times A) = \frac{1}{\ell} \int_{(-\infty,0) \times A} |a|\rho(da, dw, db),$$

for any measurable $A \subset \mathbb{R}^d \times \mathbb{R}$. Then we have

$$\begin{aligned}
g(x) &= \int a\sigma(w^\top x + b)\rho(da, dw, db) \\
&= \int_{(0,+\infty) \times \mathbb{R}^d \times \mathbb{R}} \ell \cdot \sigma(w^\top x + b) \cdot \frac{|a|}{\ell} \rho(da, dw, db) \\
&\quad + \int_{(-\infty,0) \times \mathbb{R}^d \times \mathbb{R}} (-\ell) \cdot \sigma(w^\top x + b) \cdot \frac{|a|}{\ell} \rho(da, dw, db) \\
&= \int a\sigma(w^\top x + b)\mu(da, dw, db),
\end{aligned}$$

for any $x \in \Omega$, which combined with support of $\mu$ yields that

$$\|g\|_{\mathcal{B}_R^\infty(\Omega)} \leq \ell \leq \|g\|_{\mathcal{B}_R^1(\Omega)} + \epsilon.$$

Setting $\epsilon \to 0$, we obtain that $\|g\|_{\mathcal{B}_R^\infty(\Omega)} \leq \|g\|_{\mathcal{B}_R^1(\Omega)}$. $\qquad\square$

Then we prove the approximation theorem in $H^1$ norm, i.e., Theorem 2.5.

*Proof of Theorem 2.5.* We use techniques from the proofs of [8, Theorem 1] and [2, Theorem 1, Theorem 2] to prove this theorem. According to Proposition 2.4, it holds that $g \in \mathcal{B}^2(\Omega)$ with $\|g\|_{\mathcal{B}^2(\Omega)} = \|g\|_{\mathcal{B}^1(\Omega)}$. There exists a probability measure $\rho$ supported on $\mathbb{R} \times \overline{B}_R^d \times \mathbb{R}$ such that

$$g(x) = \int a\sigma(w^\top x + b)\rho(da, dw, db), \quad x \in \Omega_0 \subset \Omega,$$

and

$$\int |a|^2 \rho(da, dw, db) \leq 2\|g\|_{\mathcal{B}^2(\Omega_0)}^2 \leq 2\|g\|_{\mathcal{B}^2(\Omega)}^2.$$

The derivatives of $g$ can also be represented in integral from,

$$\partial_j g(x) = \int a\langle w, e_j \rangle \sigma'(w^\top x + b)\rho(da, dw, db), \quad x \in \Omega_0, \ 1 \leq j \leq d,$$

where $e_j$ is a vector in $\mathbb{R}^d$ with the $j$-th entry being 1 and other entries being 0. Note that the derivative and the integral are exchangeable since

$$\int \sup_x \left| a\langle w, e_j \rangle \sigma'(w^\top x + b) \right| \rho(da, dw, db) \leq RC_1 \int |a|\rho(da, dw, db) < \infty,$$

where the last inequality holds since $\int |a|^2 \rho(da, dw, db) < \infty$ and $\rho$ is a probability measure. We sample the set of parameters $\Theta = \{a_i, w_i, b_i\}_{1 \le i \le k}$ with respect to the product measure $\rho^{\times k}$ and denote the difference between the neural network and the target function $g$ as

$$\mathcal{E}_\Theta(x) = \frac{1}{k} \sum_{i=1}^{k} a_i \sigma(w_i^\top x + b_i) - g(x).$$

Then it holds that

$$
\begin{aligned}
\mathbb{E}_{\rho^{\times k}} \|\mathcal{E}_\Theta\|^2_{L^2(\Omega_0)} &= \int_{(\mathbb{R} \times \mathbb{R}^d \times \mathbb{R})^k} \int_{\Omega_0} \left( \frac{1}{k} \sum_{i=1}^{k} a_i \sigma(w_i^\top x + b_i) - g(x) \right)^2 dx d\rho^{\times k} \\
&= \frac{1}{k^2} \int_{\Omega_0} \int_{(\mathbb{R} \times \mathbb{R}^d \times \mathbb{R})^k} \left( \sum_{i=1}^{k} \left( a_i \sigma(w_i^\top x + b_i) - g(x) \right) \right)^2 d\rho^{\times k} dx \\
&= \frac{1}{k^2} \int_{\Omega_0} \int_{(\mathbb{R} \times \mathbb{R}^d \times \mathbb{R})^k} \sum_{i=1}^{k} \left( a_i \sigma(w_i^\top x + b_i) - g(x) \right)^2 d\rho^{\times k} dx \\
&= \frac{1}{k} \int_{\Omega_0} \int_{\mathbb{R} \times \mathbb{R}^d \times \mathbb{R}} \left( a\sigma(w^\top x + b) - g(x) \right)^2 \rho(da, dw, db) dx \qquad \text{(B.1)} \\
&= \frac{1}{k} \int_{\Omega_0} \text{Var}_\rho \left( a\sigma(w^\top x + b) \right) dx \\
&\le \frac{1}{k} \int_{\Omega_0} \mathbb{E}_\rho \left[ \left( a\sigma(w^\top x + b) \right)^2 \right] dx \\
&\le \frac{C_0^2 m(\Omega_0)}{k} \mathbb{E}_\rho |a|^2 \\
&\le \frac{2 C_0^2 m(\Omega_0)}{k} \|g\|^2_{\mathcal{B}_R^2(\Omega)},
\end{aligned}
$$

and

$$
\begin{aligned}
\mathbb{E}_{\rho^{\times k}} \|\partial_j \mathcal{E}_\Theta\|^2_{L^2(\Omega_0)} &= \int_{(\mathbb{R} \times \mathbb{R}^d \times \mathbb{R})^k} \int_{\Omega_0} \left( \partial_j \left( \frac{1}{k} \sum_{i=1}^{k} a_i \sigma(w_i^\top x + b_i) \right) - \partial_j g(x) \right)^2 dx d\rho^{\times k} \\
&= \int_{(\mathbb{R} \times \mathbb{R}^d \times \mathbb{R})^k} \int_{\Omega_0} \left( \frac{1}{k} \sum_{i=1}^{k} a_i \langle w_i, e_j \rangle \sigma'(w_i^\top x + b_i) - \partial_j g(x) \right)^2 dx d\rho^{\times k} \\
&= \frac{1}{k^2} \int_{\Omega_0} \int_{(\mathbb{R} \times \mathbb{R}^d \times \mathbb{R})^k} \left( \sum_{i=1}^{k} \left( a_i \langle w_i, e_j \rangle \sigma'(w_i^\top x + b_i) - \partial_j g(x) \right) \right)^2 d\rho^{\times k} dx \\
&= \frac{1}{k^2} \int_{\Omega_0} \int_{(\mathbb{R} \times \mathbb{R}^d \times \mathbb{R})^k} \sum_{i=1}^{k} \left( a_i \langle w_i, e_j \rangle \sigma'(w_i^\top x + b_i) - \partial_j g(x) \right)^2 d\rho^{\times k} dx \\
&= \frac{1}{k} \int_{\Omega_0} \int_{\mathbb{R} \times \mathbb{R}^d \times \mathbb{R}} \left( a\langle w, e_j \rangle \sigma'(w^\top x + b) - g(x) \right)^2 \rho(da, dw, db) dx \\
&= \frac{1}{k} \int_{\Omega_0} \text{Var}_\rho \left( a\langle w, e_j \rangle \sigma'(w^\top x + b) \right) dx \\
&\le \frac{1}{k} \int_{\Omega_0} \mathbb{E}_\rho \left[ \left( a\langle w, e_j \rangle \sigma'(w^\top x + b) \right)^2 \right] dx,
\end{aligned}
$$

which then yields that

$$
\begin{aligned}
\mathbb{E}_{\rho \times k} \sum_{j=1}^{d} \|\partial_j \mathcal{E}_\Theta\|_{L^2(\Omega_0)}^2 &\leq \frac{1}{k} \int_{\Omega_0} \mathbb{E}_\rho \left[ \sum_{j=1}^{d} \left( a\langle w, e_j \rangle \sigma'(w^\top x + b) \right)^2 \right] dx \\
&\leq \frac{R^2}{k} \int_{\Omega_0} \mathbb{E}_\rho \left[ \left( a\sigma'(w^\top x + b) \right)^2 \right] dx \\
&\leq \frac{R^2 C_1^2 m(\Omega_0)}{k} \mathbb{E}_\rho |a|^2 \\
&\leq \frac{2R^2 C_1^2 m(\Omega_0)}{k} \|g\|_{\mathcal{B}_R^2(\Omega)}^2 .
\end{aligned}
\tag{B.2}
$$

Combining (B.1) with (B.2), we obtain that

$$
\mathbb{E}_{\rho \times k} \|\mathcal{E}_\Theta\|_{H^1(\Omega_0)}^2 \leq \frac{2(C_0^2 + R^2 C_1^2) m(\Omega_0) \|g\|_{\mathcal{B}_R^2(\Omega)}^2}{k}.
$$

Therefore, there exists some $\Theta = \{a_i, w_i, b_i\}_{1 \leq i \leq k}$ such that (2.3) holds. $\qquad\square$

## C Proofs for Section 3.1

In this section, we show the convergence of the iteration (3.1). We first show Proposition 3.1 that states the contraction property. Recall that the Sobolev space $H^s(\mathbb{R}^d)$ is characterized by the Fourier transform as

$$
H^s(\mathbb{R}^d) := \left\{ f \in \mathcal{S}_d' \,|\, (1 + \|\xi\|^2)^{\frac{s}{2}} \hat{f}(\xi) \in L^2(\mathbb{R}^d) \right\}, \quad s \in \mathbb{R}.
$$

Let us define the operator $P : H^s(\mathbb{R}^d) \to H^{s-2}(\mathbb{R}^d)$ by

$$
Pf = \mathcal{F}^{-1}\big( (1 + \|\xi\|^2) \hat{f}(\xi) \big).
$$

Given an index $\beta \in \mathbb{R}$, we also define the fractional power $P^\beta : H^s(\mathbb{R}^d) \to H^{s-2\beta}(\mathbb{R}^d)$ by

$$
P^\beta f = \mathcal{F}^{-1}\big( (1 + \|\xi\|^2)^\beta \hat{f}(\xi) \big).
$$

Then $P^{-1}$ is identical to $(I - \Delta)^{-1}$. It is useful to notice that

$$
\|P^{\frac{1}{2}} u\|_{L^2(\mathbb{R}^d)}^2 = \langle Pu, u \rangle_{H^{-1}(\mathbb{R}^d), H^1(\mathbb{R}^d)} = \|u\|_{H^1(\mathbb{R}^d)}^2.
\tag{C.1}
$$

We first prove some lemmas as the preparation for Proposition 3.1.

**Lemma C.1.** *Suppose that Assumption 2.1 holds. Then the linear operator*

$$
P^{-\frac{1}{2}} \mathcal{L} P^{-\frac{1}{2}} : L^2(\mathbb{R}^d) \to L^2(\mathbb{R}^d),
$$

*is bounded and self-adjoint.*

*Proof.* Consider any $u \in L^2(\mathbb{R}^d)$ with $\|u\|_{L^2(\mathbb{R}^d)} = 1$. Then $\tilde{u} = P^{-\frac{1}{2}} u \in H^1(\mathbb{R}^d)$ satisfies that $\|\tilde{u}\|_{H^1(\mathbb{R}^d)} = \|u\|_{L^2(\mathbb{R}^d)} = 1$ by (C.1). It holds that

$$
\begin{aligned}
\left\| P^{-\frac{1}{2}} \mathcal{L} P^{-\frac{1}{2}} u \right\|_{L^2(\mathbb{R}^d)} &= \sup_{\|v\|_{L^2(\mathbb{R}^d)}=1} \left\langle P^{-\frac{1}{2}} \mathcal{L} P^{-\frac{1}{2}} u, v \right\rangle_{L^2(\mathbb{R}^d)} \\
&\stackrel{\tilde{v}=P^{-\frac{1}{2}}v}{=} \sup_{\|\tilde{v}\|_{H^1(\mathbb{R}^d)}=1} \langle \mathcal{L}\tilde{u}, \tilde{v} \rangle_{H^{-1}(\mathbb{R}^d), H^1(\mathbb{R}^d)} \\
&= \sup_{\|\tilde{v}\|_{H^1(\mathbb{R}^d)}=1} \int_{\mathbb{R}^d} (A\nabla\tilde{u} \cdot \nabla\tilde{v} + c\tilde{u}\tilde{v}) dx \\
&\leq \sup_{\|\tilde{v}\|_{H^1(\mathbb{R}^d)}=1} \max\{a_{\max}, c_{\max}\} \|\tilde{u}\|_{H^1(\mathbb{R}^d)} \|\tilde{v}\|_{H^1(\mathbb{R}^d)} \\
&= \lambda_{\max}.
\end{aligned}
$$

Therefore, $P^{-\frac{1}{2}}\mathcal{L}P^{-\frac{1}{2}}$ is bounded on $L^2(\mathbb{R}^d)$.

For any $u, v \in L^2(\mathbb{R}^d)$ with $\tilde{u} = P^{-\frac{1}{2}}u$ and $\tilde{v} = P^{-\frac{1}{2}}v$, by the symmetry of $A$, we have that

$$\left\langle P^{-\frac{1}{2}}\mathcal{L}P^{-\frac{1}{2}}u, v \right\rangle_{L^2(\mathbb{R}^d)} = \langle \mathcal{L}\tilde{u}, \tilde{v} \rangle_{H^{-1}(\mathbb{R}^d), H^1(\mathbb{R}^d)}$$

$$= \int_{\mathbb{R}^d} (A\nabla\tilde{u} \cdot \nabla\tilde{v} + c\tilde{u}\tilde{v})dx$$

$$= \int_{\mathbb{R}^d} (\nabla\tilde{u} \cdot A\nabla\tilde{v} + c\tilde{u}\tilde{v})dx$$

$$= \langle \mathcal{L}\tilde{v}, \tilde{u} \rangle_{H^{-1}(\mathbb{R}^d), H^1(\mathbb{R}^d)}$$

$$= \left\langle u, P^{-\frac{1}{2}}\mathcal{L}P^{-\frac{1}{2}}v \right\rangle_{L^2(\mathbb{R}^d)},$$

which implies that $P^{-\frac{1}{2}}\mathcal{L}P^{-\frac{1}{2}}$ is self-adjoint on $L^2(\mathbb{R}^d)$. $\qquad\square$

The following lemma will also be useful. Let $T$ be a bounded linear operator on a Hilbert space $H$. Denote by $\sigma(T)$ the set of spectrum of $T$ and by $r(T) := \sup\{|\lambda| \,|\, \lambda \in \sigma(T)\}$ the spectrum radius. Define the numerical range $\mathcal{W}(T)$ of $T$ by

$$\mathcal{W}(T) := \{\langle Th, h\rangle, \|h\| = 1\}.$$

The numerical radius is defined as $w(T) := \sup\{|\lambda| \,|\, \lambda \in \mathcal{W}(T)\}$.

**Lemma C.2.** *Let $T$ be a bounded linear operator on a Hilbert space $H$. Then*

$$r(T) \le w(T).$$

*Proof.* The proof follows directly from the fact that

$$\sigma(T) \subset \overline{\mathcal{W}(T)}.$$

See e.g. [21, Theorem 6.2.1] for the statement and proof of the above. $\qquad\square$

We then prove Proposition 3.1.

*Proof of Proposition 3.1.* First it follows from (C.1) that

$$\left\|(I - \alpha(I-\Delta)^{-1}\mathcal{L})u\right\|^2_{H^1(\mathbb{R}^d)} = \left\|(I - \alpha P^{-1}\mathcal{L})u\right\|^2_{H^1(\mathbb{R}^d)}$$

$$= \left\|P^{\frac{1}{2}}(I - \alpha P^{-1}\mathcal{L})u\right\|^2_{L^2(\mathbb{R}^d)}$$

$$= \left\|P^{\frac{1}{2}}(I - \alpha P^{-1}\mathcal{L})P^{-\frac{1}{2}}P^{\frac{1}{2}}u\right\|^2_{L^2(\mathbb{R}^d)}$$

$$\le \left\|P^{\frac{1}{2}}(I - \alpha P^{-1}\mathcal{L})P^{-\frac{1}{2}}\right\|^2_{L^2(\mathbb{R}^d)\to L^2(\mathbb{R}^d)} \left\|P^{\frac{1}{2}}u\right\|^2_{L^2(\mathbb{R}^d)}$$

$$= \left\|I - \alpha P^{-\frac{1}{2}}\mathcal{L}P^{-\frac{1}{2}}\right\|^2_{L^2(\mathbb{R}^d)\to L^2(\mathbb{R}^d)} \|u\|^2_{H^1(\mathbb{R}^d)}.$$

Notice that the operator $I - \alpha P^{-\frac{1}{2}}\mathcal{L}P^{-\frac{1}{2}}$ is bounded and self-adjoint on $L^2(\mathbb{R}^d)$ by Lemma C.1. Therefore $\left\|I - \alpha P^{-\frac{1}{2}}\mathcal{L}P^{-\frac{1}{2}}\right\|_{L^2(\mathbb{R}^d)\to L^2(\mathbb{R}^d)} = r(I - \alpha P^{-\frac{1}{2}}\mathcal{L}P^{-\frac{1}{2}})$. In addition, thanks to Lemma C.2,

$$r(I - \alpha P^{-\frac{1}{2}}\mathcal{L}P^{-\frac{1}{2}}) \le w(I - \alpha P^{-\frac{1}{2}}\mathcal{L}P^{-\frac{1}{2}}).$$

By the definition of numerical radius and the identity (C.1), one has that

$$w(I - \alpha P^{-\frac{1}{2}}LP^{-\frac{1}{2}}) = \sup_{\|u\|_{L^2(\mathbb{R}^d)}=1} \left| \left\langle (I - \alpha P^{-\frac{1}{2}}\mathcal{L}P^{-\frac{1}{2}})u, u \right\rangle_{L^2(\mathbb{R}^d)} \right|$$

$$\overset{\tilde{u}=P^{-\frac{1}{2}}u}{=} \sup_{\|\tilde{u}\|_{H^1(\mathbb{R}^d)}=1} \left| 1 - \alpha\langle \mathcal{L}\tilde{u}, \tilde{u}\rangle_{H^{-1}(\mathbb{R}^d), H^1(\mathbb{R}^d)} \right|$$

$$= \sup_{\|\tilde{u}\|_{H^1(\mathbb{R}^d)}=1} \left| 1 - \alpha \int_{\mathbb{R}^d} (A\nabla\tilde{u} \cdot \nabla\tilde{u} + c|\tilde{u}|^2)dx \right|.$$

Moreover, thanks to the positivity and boundedness assumptions on $A$ and $c$, we have for any $\tilde{u}$ with $\|\tilde{u}\|_{H^1(\mathbb{R}^d)} = 1$,

$$\lambda_{\min} = \min\{a_{\min}, c_{\min}\} \leq \int_{\mathbb{R}^d} \nabla \tilde{u} \cdot A \nabla \tilde{u} + c|\tilde{u}|^2 dx \leq \max\{a_{\max}, c_{\max}\} = \lambda_{\max}.$$

Therefore we have obtained that

$$w(I - \alpha P^{-\frac{1}{2}} \mathcal{L} P^{-\frac{1}{2}}) \leq \Lambda(\alpha).$$

Combining the estimates above finishes the proof of the first inequality in (3.2). Finally, the second inequality (3.3) follows by optimizing the function $\Lambda(\alpha)$ with respect to $\alpha > 0$. In fact it is not hard to verify that

$$\inf_{\alpha > 0} \Lambda(\alpha) = \Lambda(\alpha_*) = \frac{\lambda_{\max} - \lambda_{\min}}{\lambda_{\max} + \lambda_{\min}},$$

where $\alpha_* = \frac{2}{\lambda_{\min} + \lambda_{\max}}$. $\qquad\square$

*Proof of Corollary 3.2.* It follows from (3.1) and Proposition 3.1 that

$$\|u_{t+1} - u^*\|_{H^1(\mathbb{R}^d)} = \left\| \left(I - \alpha_*(I - \Delta)^{-1} \mathcal{L}\right)(u_t - u^*) \right\|_{H^1(\mathbb{R}^d)}$$

$$\leq \frac{\lambda_{\max} - \lambda_{\min}}{\lambda_{\max} + \lambda_{\min}} \|u_t - u^*\|_{H^1(\mathbb{R}^d)},$$

which then implies that

$$\|u_T - u^*\|_{H^1(\mathbb{R}^n)} \leq \left( \frac{\lambda_{\max} - \lambda_{\min}}{\lambda_{\max} + \lambda_{\min}} \right)^T \|u_0 - u^*\|_{H^1(\mathbb{R}^d)} \leq \epsilon,$$

for $T$ satisfying (3.2). $\qquad\square$

# D  Proofs for Section 3.2

In this section, we give proofs for Lemma 3.3, Lemma 3.4, Lemma 3.5, and Lemma 3.6. These lemmas show that the updating rule (3.1) keeps the iterates $\{u_t\}_{t\in\mathbb{N}}$ staying in the Barron space and estimate the amplification of Barron norm after performing (3.1).

*Proof of Lemma 3.3.* (i) (Addition) Let $\epsilon > 0$ be fixed. For any $i \in \{1, 2, \ldots, k\}$, there exists a probability measure $\rho_i$ supported on $\mathbb{R} \times \overline{B}_{R_i}^d \times \mathbb{R}$ such that

$$g_i(x) = \int a\sigma(w^\top x + b)\rho_i(da, dw, db), \quad x \in \mathbb{R}^d,$$

and that

$$\int |a|\rho_i(da, dw, db) \leq \|g_i\|_{\mathcal{B}_{R_i}^1(\mathbb{R}^d)} + \epsilon.$$

We have

$$(g_1 + \cdots + g_k)(x) = \int a\sigma(w^\top x + b)(\rho_1 + \cdots + \rho_k)(da, dw, db)$$

$$= \int ka\sigma(w^\top x + b)\frac{\rho_1 + \cdots + \rho_k}{k}(da, dw, db).$$

Consider a function $F : \mathbb{R} \times \mathbb{R}^d \times \mathbb{R} \to \mathbb{R} \times \mathbb{R}^d \times \mathbb{R}$, $(a, w, b) \mapsto (ka, w, b)$ and the corresponding push-forward measure $\rho = F_* \frac{\rho_1 + \cdots + \rho_k}{k}$. Noticing that $\rho$ is supported on $\mathbb{R} \times \overline{B}_R^d \times \mathbb{R}$, where $R = \max_{1 \leq i \leq k} R_i$, and that

$$(g_1 + \cdots + g_k)(x) = \int a\sigma(w^\top x + b)\rho(da, dw, db), \quad x \in \mathbb{R}^d,$$

we obtain that

$$\|g_1 + \cdots + g_k\|_{\mathcal{B}_R^1(\mathbb{R}^d)} \leq \int |a|\rho(da, dw, db)$$

$$= \int |ka| \frac{\rho_1 + \cdots + \rho_k}{k} (da, dw, db)$$

$$= \sum_{i=1}^{k} \int |a| \rho_i (da, dw, db)$$

$$\leq \sum_{1 \leq i \leq k} \|g_i\|_{\mathcal{B}^1_{R_i}(\mathbb{R}^d)} + k\epsilon.$$

Then we can conclude that $\|g_1 + \cdots + g_k\|_{\mathcal{B}^1_R(\mathbb{R}^d)} \leq \sum_{1 \leq i \leq k} \|g_i\|_{\mathcal{B}^1_{R_i}(\mathbb{R}^d)}$ by setting $\epsilon \to 0$.

(ii) (Scalar multiplication) The result is trivial if $\lambda = 0$. We then consider $\lambda \neq 0$. For any $\epsilon > 0$, there exists a probability measure $\rho$ supported on $\mathbb{R} \times \overline{B}^d_R \times \mathbb{R}$ such that

$$g(x) = \int a\sigma(w^\top x + b)\rho(da, dw, db), \quad x \in \mathbb{R}^d,$$

and that

$$\int |a|\rho(da, dw, db) \leq \|g\|_{\mathcal{B}^1_R(\mathbb{R}^d)} + \epsilon.$$

Then it holds that

$$(\lambda g)(x) = \int \lambda a\sigma(w^\top x + b)\rho(da, dw, db) = \int a\sigma(w^\top x + b)\rho'(da, dw, db),$$

where $\rho' = F_*\rho$ is the push-forward measure and $F : \mathbb{R} \times \mathbb{R}^d \times \mathbb{R} \to \mathbb{R} \times \mathbb{R}^d \times \mathbb{R}$, $(a, w, b) \mapsto (\lambda a, w, b)$. Since $\rho'$ is supported on $\mathbb{R} \times \overline{B}^d_R \times \mathbb{R}$, we get that

$$\|\lambda g\|_{\mathcal{B}^1_R(\mathbb{R}^d)} \leq \int |a|\rho'(da, dw, db) = \int |\lambda a|\rho(da, dw, db) \leq |\lambda| \|g\|_{\mathcal{B}^1_R(\mathbb{R}^d)} + |\lambda|\epsilon,$$

which implies that $\|\lambda g\|_{\mathcal{B}^1_R(\mathbb{R}^d)} \leq |\lambda| \|g\|_{\mathcal{B}^1_R(\mathbb{R}^d)}$ by setting $\epsilon \to 0$. Furthermore, we have that

$$\|\lambda g\|_{\mathcal{B}^1_R(\mathbb{R}^d)} \leq |\lambda| \|g\|_{\mathcal{B}^1_R(\mathbb{R}^d)} = |\lambda| \left\|\lambda^{-1} \cdot \lambda g\right\|_{\mathcal{B}^1_R(\mathbb{R}^d)} \leq |\lambda \cdot \lambda^{-1}| \|\lambda g\|_{\mathcal{B}^1_R(\mathbb{R}^d)} = \|\lambda g\|_{\mathcal{B}^1_R(\mathbb{R}^d)}.$$

Thus, the equalities must hold and $\|\lambda g\|_{\mathcal{B}^1_R(\mathbb{R}^d)} = |\lambda| \|g\|_{\mathcal{B}^1_R(\mathbb{R}^d)}$

(iii) (Product) Fix $\epsilon > 0$. For $i \in \{1, 2\}$, there exists a probability $\rho_i$ supported on $\mathbb{R} \times \overline{B}^d_{R_i} \times \mathbb{R}$ such that

$$g_i(x) = \int a\sigma(w^\top x + b)\rho_i(da, dw, db), \quad x \in \mathbb{R}^d,$$

and that

$$\int |a|\rho_i(da, dw, db) \leq \|g_i\|_{\mathcal{B}^1_{R_i}(\mathbb{R}^d)} + \epsilon.$$

According to Assumption 2.7, there exists a probability measure $\mu$ supported on $\mathbb{R} \times \overline{B}^2_{R_m} \times \mathbb{R}$ such that

$$\sigma(y_1)\sigma(y_2) = \int \gamma\sigma(\xi_1 y_1 + \xi_2 y_2 + \eta)\mu(d\gamma, d\xi, d\eta), \quad y_1, y_2 \in \mathbb{R},$$

where $\xi = (\xi_1, \xi_2)^\top \in \mathbb{R}^2$, and

$$\int |\gamma|\mu(d\gamma, d\xi, d\eta) \leq \ell_m + \epsilon.$$

Recall that $\sup_{y \in \mathbb{R}} |\sigma(y)| < \infty$. By Fubini's theorem, it holds for any $x \in \mathbb{R}^d$ that

$$g_1(x)g_2(x) = \int a_1\sigma(w_1^\top x + b_1)\rho_1(da_1, dw_1, db_1) \int a_2\sigma(w_2^\top x + b_2)\rho_2(da_2, dw_2, db_2)$$

$$= \int a_1 a_2 \sigma(w_1^\top x + b_1)\sigma(w_2^\top x + b_2)\rho_1 \times \rho_2(da_1, dw_1, db_1, da_2, dw_2, db_2)$$

$$= \int a_1 a_2 \int \gamma\sigma\left(\xi_1(w_1^\top x + b_1) + \xi_2(w_2^\top x + b_2) + \eta\right)\mu(d\gamma, d\xi, d\eta)$$

$$\rho_1 \times \rho_2(da_1, dw_1, db_1, da_2, dw_2, db_2)$$

$$= \int a_1 a_2 \gamma \sigma \left( (\xi_1 w_1 + \xi_2 w_2)^\top x + \xi_1 b_1 + \xi_2 b_2 + \eta \right)$$

$$\rho_1 \times \rho_2 \times \mu(da_1, dw_1, db_1, da_2, dw_2, db_2, d\gamma, d\xi, d\eta).$$

Consider a function

$$F : \mathbb{R} \times \mathbb{R}^d \times \mathbb{R} \times \mathbb{R} \times \mathbb{R}^d \times \mathbb{R} \times \mathbb{R}^2 \times \mathbb{R} \to \mathbb{R} \times \mathbb{R}^d \times \mathbb{R},$$

$$(a_1, w_1, b_1, a_2, w_2, b_2, \gamma, \xi, \eta) \qquad \mapsto \quad (a', w', b'),$$

where

$$\begin{cases} a' = a_1 a_2 \gamma, \\ w' = \xi_1 w_1 + \xi_2 w_2, \\ b' = \xi_1 b_1 + \xi_2 b_2 + \eta. \end{cases}$$

The push-forward measure $\rho' = F_*(\rho_1 \times \rho_2 \times \mu)$ is supported on $\mathbb{R} \times \overline{B}_R^d \times \mathbb{R}$ where $R = R_m(R_1 + R_2)$ and it holds that

$$g_1(x)g_2(x) = \int a\sigma(w^\top x + b)\rho'(da, dw, db).$$

Therefore, we have

$$\|g_1 g_2\|_{\mathcal{B}_R^1(\mathbb{R}^d)} \leq \int |a|\rho'(da, dw, db)$$

$$= \int |a_1 a_2 \gamma| \rho_1 \times \rho_2 \times \mu(da_1, dw_1, db_1, da_2, dw_2, db_2, d\gamma, d\xi, d\eta)$$

$$= \int |\gamma| \mu(d\gamma, d\xi, d\eta) \int |a_1| \rho_1(da_1, dw_1, db_1) \int |a_2| \rho_2(da_2, dw_2, db_2)$$

$$\leq (\ell_m + \epsilon) \left( \|g\|_{\mathcal{B}_{R_1}^1(\mathbb{R}^d)} + \epsilon \right) \left( \|g\|_{\mathcal{B}_{R_2}^1(\mathbb{R}^d)} + \epsilon \right),$$

which then implies that $\|g_1 g_2\|_{\mathcal{B}_R^1(\mathbb{R}^d)} \leq \ell_m \|g\|_{\mathcal{B}_{R_1}^1(\mathbb{R}^d)} \|g\|_{\mathcal{B}_{R_2}^1(\mathbb{R}^d)}$ as $\epsilon \to 0$.

(iv) (Derivatives) For any $\epsilon > 0$, there exists a probability measure $\rho$ supported on $\mathbb{R} \times \overline{B}_R^d \times \mathbb{R}$ such that

$$g(x) = \int a\sigma(w^\top x + b)\rho(da, dw, db), \quad x \in \mathbb{R}^d,$$

and that

$$\int |a|\rho(da, dw, db) \leq \|g\|_{\mathcal{B}_R^1(\mathbb{R}^d)} + \epsilon.$$

According to Assumption 2.8, there exist probability measures $\mu_1$ and $\mu_2$, supported on $\mathbb{R} \times \overline{B}_{R_{d,1}}^1 \times \mathbb{R}$ and $\mathbb{R} \times \overline{B}_{R_{d,2}}^1 \times \mathbb{R}$ respectively, such that

$$\sigma'(y) = \int \gamma \sigma(\xi y + \eta)\mu_1(d\gamma, d\xi, d\eta), \quad y \in \mathbb{R},$$

$$\sigma''(y) = \int \gamma \sigma(\xi y + \eta)\mu_2(d\gamma, d\xi, d\eta), \quad y \in \mathbb{R},$$

and

$$\int |\gamma|\mu_1(d\gamma, d\xi, d\eta) \leq \ell_{d,1} + \epsilon, \quad \int |\gamma|\mu_1(d\gamma, d\xi, d\eta) \leq \ell_{d,2} + \epsilon.$$

Recall that $\sup_{y \in \mathbb{R}} |\sigma(y)| < \infty$, $\sup_{y \in \mathbb{R}} |\sigma'(y)| < \infty$, $\sup_{y \in \mathbb{R}} |\sigma''(y)| < \infty$, and $\|w\| \leq R$ for $\rho$-a.e. $(a, w, b)$. It holds that

$$\partial_i g(x) = \int a\langle w, e_i \rangle \sigma'(w^\top x + b)\rho(da, dw, db)$$

$$= \int a\langle w, e_i \rangle \int \gamma \sigma(\xi(w^\top x + b) + \eta)\mu_1(d\gamma, d\xi, d\eta)\rho(da, dw, db)$$

$$= \int \gamma a \langle w, e_i \rangle \sigma((\xi w)^\top x + \xi b + \eta) \rho \times \mu_1(da, dw, db, d\gamma, d\xi, d\eta)$$

$$= \int a \sigma(w^\top x + b) \rho_1'(da, dw, db),$$

and that

$$\partial_i \partial_j g(x) = \int a \langle w, e_i \rangle \langle w, e_j \rangle \sigma''(w^\top x + b) \rho(da, dw, db)$$

$$= \int a \langle w, e_i \rangle \langle w, e_j \rangle \int \gamma \sigma(\xi(w^\top x + b) + \eta) \mu_2(d\gamma, d\xi, d\eta) \rho(da, dw, db)$$

$$= \int \gamma a \langle w, e_i \rangle \langle w, e_j \rangle \sigma((\xi w)^\top x + \xi b + \eta) \rho \times \mu_2(da, dw, db, d\gamma, d\xi, d\eta)$$

$$= \int a \sigma(w^\top x + b) \rho_2'(da, dw, db),$$

where $\rho_1' = F_*(\rho \times \mu_1)$ and $\rho_2' = G_*(\rho \times \mu_1)$ with $F(a, w, b, \gamma, \xi, \eta) = (\gamma a \langle w, e_i \rangle, \xi w, \xi b + \eta)$ and $G(a, w, b, \gamma, \xi, \eta) = (\gamma a \langle w, e_i \rangle \langle w, e_j \rangle, \xi w, \xi b + \eta)$. Note that $\rho_1'$ and $\rho_2'$ are supported on $\mathbb{R} \times \overline{B}^1_{R_{d,1}R} \times \mathbb{R}$ and $\mathbb{R} \times \overline{B}^1_{R_{d,1}R} \times \mathbb{R}$ respectively. Therefore, we obtain that

$$\|\partial_i g\|_{\mathcal{B}^1_{R_{d,1}R}(\mathbb{R}^d)} \leq \int |a| \rho_1'(da, dw, db)$$

$$= \int |\gamma a \langle w, e_i \rangle| \rho \times \mu_1(da, dw, db, d\gamma, d\xi, d\eta)$$

$$\leq R \int |\gamma| \mu_1(d\gamma, d\xi, d\eta) \int |a| \rho(da, dw, db)$$

$$\leq R(\ell_{d,1} + \epsilon) \left( \|g\|_{\mathcal{B}^1_R(\mathbb{R}^d)} + \epsilon \right),$$

and similarly,

$$\|\partial_i \partial_j g\|_{\mathcal{B}^1_{R_{d,1}R}(\mathbb{R}^d)} \leq \int |a| \rho_2'(da, dw, db)$$

$$= \int |\gamma a \langle w, e_i \rangle \langle w, e_j \rangle| \rho \times \mu_2(da, dw, db, d\gamma, d\xi, d\eta)$$

$$\leq R^2 \int |\gamma| \mu_2(d\gamma, d\xi, d\eta) \int |a| \rho(da, dw, db)$$

$$= R^2(\ell_{d,2} + \epsilon) \left( \|g\|_{\mathcal{B}^1_R(\mathbb{R}^d)} + \epsilon \right).$$

Then we can obtain the desired estimates by letting $\epsilon \to 0$. $\qquad\square$

*Proof of Lemma 3.4.* This proof is modified from [9]. Consider the one-dimensional case $d = 1$ first. The Green's function $G(x)$ for the screened Poisson equation

$$\left( I - \frac{d^2}{dx^2} \right) u = g, \quad x \in \mathbb{R},$$

can be explicitly computed as

$$G(x) = \mathcal{F}^{-1} \left( \frac{1}{1 + \xi^2} \mathcal{F} \delta_0 \right) = \mathcal{F}^{-1} \left( \frac{1}{1 + \xi^2} \right) = \frac{1}{2} e^{-|x|},$$

which leads to $\int_{\mathbb{R}} |G(x)| dx = 1$. For any $\epsilon > 0$, there exists a probability measure $\rho$ supported on $\mathbb{R} \times \overline{B}_R \times \mathbb{R}$ such that

$$g(x) = \int a \sigma(wx + b) \rho(da, dw, db), \quad x \in \mathbb{R},$$

and that

$$\int |a| \rho(da, dw, db) \leq \|g\|_{\mathcal{B}^1_R(\mathbb{R})} + \epsilon.$$

It holds that

$$\left(\left(I - \frac{d^2}{dx^2}\right)^{-1} g\right)(x) = \int G(y)g(x-y)dy$$

$$= \int G(y) \int a\sigma(w(x-y)+b)\rho(da,dw,db)dy$$

$$= \int G(y)a\sigma(wx - wy + b)\rho(da,dw,db)dy$$

$$= \int a\sigma(wx+b)\rho'(da,dw,db),$$

where $\rho' = F_*(\rho \times m)$ with $m$ being the Lebesgue measure and $F : \mathbb{R} \times \mathbb{R} \times \mathbb{R} \times \mathbb{R} \to \mathbb{R} \times \mathbb{R} \times \mathbb{R}$, $(a,w,b,y) \mapsto (G(y)a,w,b-wy)$. Then $\rho'$ is supported on $\mathbb{R} \times \overline{B}_R \times \mathbb{R}$ and by Fubini's theorem,

$$\left\|\left(I - \frac{d^2}{dx^2}\right)^{-1} g\right\|_{\mathcal{B}_R^1(\mathbb{R})} \leq \int |a|\rho'(da,dw,db)$$

$$= \int |G(y)a|\rho(da,dw,db)dy$$

$$= \int |G(y)|dy \int |a|\rho(da,dw,db)$$

$$\leq \|g\|_{\mathcal{B}_R^1(\mathbb{R})} + \epsilon,$$

which leads to $\left\|\left(I - \frac{d^2}{dx^2}\right)^{-1} g\right\|_{\mathcal{B}_R^1(\mathbb{R})} \leq \|g\|_{\mathcal{B}_R^1(\mathbb{R})}$ as $\epsilon \to 0$. For a general dimension $d \geq 1$, since the operator $I - \Delta$ and the Barron norm are invariant under any orthogonal transformation, for any $w \in \overline{B}_R^d$ and $b \in \mathbb{R}$, by the analysis for $d = 1$, it holds that

$$\left\|(I - \Delta)^{-1}\sigma(w^\top \cdot + b)\right\|_{\mathcal{B}_R^1(\mathbb{R}^d)} \leq \left\|\sigma(w^\top \cdot + b)\right\|_{\mathcal{B}_R^1(\mathbb{R}^d)} \leq 1.$$

For any $\epsilon > 0$, there exists a probability measure $\rho$ supported on $\mathbb{R} \times \overline{B}_R^d \times \mathbb{R}$ such that

$$g(x) = \int a\sigma(w^\top x + b)\rho(da,dw,db), \quad x \in \mathbb{R}^d,$$

and that

$$\int |a|\rho(da,dw,db) \leq \|g\|_{\mathcal{B}_R^1(\mathbb{R}^d)} + \epsilon.$$

Therefore, by the Jensen's inequality for expectation and the convexity of the Barron norm, one has that

$$\left\|(I - \Delta)^{-1}g\right\|_{\mathcal{B}_R^1(\mathbb{R}^d)} = \left\|\int a(I - \Delta)^{-1}\sigma(w^\top \cdot + b)\rho(da,dw,db)\right\|_{\mathcal{B}_R^1(\mathbb{R}^d)}$$

$$\leq \int |a| \left\|(I - \Delta)^{-1}\sigma(w^\top \cdot + b)\right\|_{\mathcal{B}_R^1(\mathbb{R}^d)} \rho(da,dw,db)$$

$$\leq \int |a|\rho(da,dw,db)$$

$$\leq \|g\|_{\mathcal{B}_R^1(\mathbb{R}^d)} + \epsilon,$$

which yields that $\left\|(I - \Delta)^{-1}g\right\|_{\mathcal{B}_R^1(\mathbb{R})} \leq \|g\|_{\mathcal{B}_R^1(\mathbb{R})}$ as $\epsilon \to 0$. $\square$

*Proof of Lemma 3.5.* According to Lemma 3.3 (iv), we have

$$\|\partial_i A_{ij}\|_{\mathcal{B}_{R_{d,1}R_A}^1(\mathbb{R}^d)} \leq \ell_{d,1}R_A \|A_{i,j}\|_{\mathcal{B}_{R_A}^1(\mathbb{R}^d)},$$

$$\|\partial_j u_t\|_{\mathcal{B}_{R_{d,1}R_{u,t}}^1(\mathbb{R}^d)} \leq \ell_{d,1}R_{u,t} \|u_t\|_{\mathcal{B}_{R_{u,t}}^1(\mathbb{R}^d)},$$

and

$$\|\partial_{ij}u_t\|_{\mathcal{B}^1_{R_{d,2}R_{u,t}}(\mathbb{R}^d)} \leq \ell_{d,2}R_{u,t}^2\|u_t\|_{\mathcal{B}^1_{R_{u,t}}(\mathbb{R}^d)},$$

for any $1 \leq i,j \leq d$. Then applying Lemma 3.3 (iii), we obtain that

$$\|cu_t\|_{\mathcal{B}^1_{R_m(R_{u,t}+R_c)}(\mathbb{R}^d)} \leq \ell_m\|u_t\|_{\mathcal{B}^1_{R_{u,t}}(\mathbb{R}^d)}\|c\|_{\mathcal{B}^1_{R_c}(\mathbb{R}^d)} = \ell_m\ell_c\|u_t\|_{\mathcal{B}^1_{R_{u,t}}(\mathbb{R}^d)},$$

$$\|\partial_i A_{ij}\partial_j u_t\|_{\mathcal{B}^1_{R_m R_{d,1}(R_{u,t}+R_A)}(\mathbb{R}^d)} \leq \ell_m\ell_{d,1}^2 R_A R_{u,t}\|A_{i,j}\|_{\mathcal{B}^1_{R_A}(\mathbb{R}^d)}\|u_t\|_{\mathcal{B}^1_{R_{u,t}}(\mathbb{R}^d)}$$
$$\leq \ell_m\ell_{d,1}^2\ell_A R_A R_{u,t}\|u_t\|_{\mathcal{B}^1_{R_{u,t}}(\mathbb{R}^d)},$$

and

$$\|A_{ij}\partial_{ij}u_t\|_{\mathcal{B}^1_{R_m(R_{d,2}R_{u,t}+R_A)}(\mathbb{R}^d)} \leq \ell_m\ell_{d,2}R_{u,t}^2\|A_{ij}\|_{\mathcal{B}^1_{R_A}}\|u_t\|_{\mathcal{B}^1_{R_{u,t}}(\mathbb{R}^d)}$$
$$\leq \ell_m\ell_{d,2}\ell_A R_{u,t}^2\|u_t\|_{\mathcal{B}^1_{R_{u,t}}(\mathbb{R}^d)},$$

for any $1 \leq i,j \leq d$. Therefore, one can estimate the Barron norm of $v_t$ by Lemma 3.3 (i):

$$\|v_t\|_{\mathcal{B}^1_{R_{v,t}}(\mathbb{R}^d)} \leq \left(\ell_m\ell_A(\ell_{d,1}^2 R_A R_{u,t} + \ell_{d,2}R_{u,t}^2)d^2 + \ell_m\ell_c\right)\|u_t\|_{\mathcal{B}^1_{R_{u,t}}(\mathbb{R}^d)} + \ell_f,$$

where

$$R_{v,t} = \max\{R_m R_{d,1}(R_{u,t}+R_A), R_m(R_{d,2}R_{u,t}+R_A), R_m(R_{u,t}+R_c), R_f\}.$$

Then using Lemma 3.4, Lemma 3.3 (ii), and Lemma 3.3 (i), we can finally conclude (3.5). $\qquad\square$

*Proof of Lemma 3.6.* Consider any $\epsilon > 0$. There exists a probability measure $\rho$ supported on $\mathbb{R} \times \overline{B}^d_{R_{u,t}} \times \mathbb{R}$, such that

$$u_t(x) = \int a\cos(w^\top x + b)\rho(da, dw, db), \quad x \in \mathbb{R}^d,$$

and

$$\int |a|\rho(da, dw, db) \leq \|u_t\|_{\mathcal{B}^1_{R_t}(\mathbb{R}^d)} + \epsilon.$$

Let us suppose that

$$A_{ij}(x) = \int a_{A,ij}\cos(w_{A,ij}^\top x + b_{A,ij})\rho_{A,ij}(da_{A,ij}, dw_{A,ij}, db_{A,ij}), \quad x \in \mathbb{R}^d,$$

$$c(x) = \int a_c\cos(w_c^\top x + b_c)\rho_c(da_c, dw_c, db_c), \quad x \in \mathbb{R}^d,$$

and

$$f(x) = \int a_f\cos(w_f^\top x + b_f)\rho_f(da_f, dw_f, db_f), \quad x \in \mathbb{R}^d,$$

where the probability measures $\rho_{A,ij}$, $\rho_c$, and $\rho_f$ are supported on $\mathbb{R} \times \overline{B}^d_{R_A} \times \mathbb{R}$, $\mathbb{R} \times \overline{B}^d_{R_c} \times \mathbb{R}$, and $\mathbb{R} \times \overline{B}^d_{R_f} \times \mathbb{R}$, respectively, and satisfy that

$$\int |a_{A,ij}|\rho_{A,ij}(da_{A,ij}, dw_{A,ij}, db_{A,ij}) \leq \ell_A + \epsilon,$$

$$\int |a_c|\rho_c(da_c, dw_c, db_c) \leq \ell_c + \epsilon,$$

and

$$\int |a_f|\rho_f(da_f, dw_f, db_f) \leq \ell_f + \epsilon.$$

Then it holds that

$$
\begin{aligned}
v_t(x) =& \mathcal{L}u_t - f \\
=& -\sum_{i,j}(\partial_i A_{ij}\partial_j u_t + A_{ij}\partial_{ij}u_t) + cu_t - f \\
=& -\sum_{i,j}\left(\int a_{A,ij}\langle w_{A,ij}, e_i\rangle \cos\left(w_{A,ij}^\top x + b_{A,ij} + \frac{\pi}{2}\right) d\rho_{A,ij}\right. \\
& \qquad\qquad \cdot \int a\langle a, w_j\rangle \cos\left(w^\top x + b + \frac{\pi}{2}\right) d\rho \\
& \qquad + \left.\int a_{A,ij}\cos(w_{A,ij}^\top x + b_{A,ij})d\rho_{A,ij}\int a\langle w, e_i\rangle\langle w, e_j\rangle \cos(w^\top x + b + \pi)d\rho\right) \\
& + \int a_c\cos(w_c^\top x + b_c)d\rho_c\int a\cos(w^\top x + b)d\rho \\
& + f(x) = \int a_f\cos(w_f^\top x + b_f)d\rho_f \\
=& -\sum_{i,j}\left(\frac{a_{A,ij}a\langle w_{A,ij}, e_i\rangle\langle w, e_j\rangle}{2}\cos\left((w_{A,ij}+w)^\top x + b_{A,ij} + b + \pi\right) d\rho_{A,ij}\times d\rho\right. \\
& \qquad + \frac{a_{A,ij}a\langle w_{A,ij}, e_i\rangle\langle w, e_j\rangle}{2}\cos\left((w_{A,ij}-w)^\top x + b_{A,ij} - b\right) d\rho_{A,ij}\times d\rho \\
& \qquad + \frac{a_{A,ij}a\langle w, e_i\rangle\langle w, e_j\rangle}{2}\cos\left((w_{A,ij}+w)^\top x + b_{A,ij} + b + \pi\right) d\rho_{A,ij}\times d\rho \\
& \qquad + \left.\frac{a_{A,ij}a\langle w, e_i\rangle\langle w, e_j\rangle}{2}\cos\left((w_{A,ij}-w)^\top x + b_{A,ij} - b - \pi\right) d\rho_{A,ij}\times d\rho\right) \\
& + \frac{a_c a}{2}\cos\left((w_c+w)^\top x + b_c + b\right) d\rho_c\times d\rho \\
& + \frac{a_c a}{2}\cos\left((w_c-w)^\top x + b_c - b\right) d\rho_c\times d\rho \\
& - a_f\cos(w_f^\top x + b_f)d\rho_f.
\end{aligned}
$$

Let us denote

$$
\begin{aligned}
\tilde{v}_t(x) =& -\sum_{i,j}\left(\frac{a_{A,ij}a\langle w_{A,ij}, e_i\rangle\langle w, e_j\rangle}{2(1 + \|w_{A,ij}+w\|^2)}\cos\left((w_{A,ij}+w)^\top x + b_{A,ij} + b + \pi\right) d\rho_{A,ij}\times d\rho\right. \\
& \qquad + \frac{a_{A,ij}a\langle w_{A,ij}, e_i\rangle\langle w, e_j\rangle}{2(1 + \|w_{A,ij}-w\|^2)}\cos\left((w_{A,ij}-w)^\top x + b_{A,ij} - b\right) d\rho_{A,ij}\times d\rho \\
& \qquad + \frac{a_{A,ij}a\langle w, e_i\rangle\langle w, e_j\rangle}{2(1 + \|w_{A,ij}+w\|^2)}\cos\left((w_{A,ij}+w)^\top x + b_{A,ij} + b + \pi\right) d\rho_{A,ij}\times d\rho \\
& \qquad + \left.\frac{a_{A,ij}a\langle w, e_i\rangle\langle w, e_j\rangle}{2(1 + \|w_{A,ij}-w\|^2)}\cos\left((w_{A,ij}-w)^\top x + b_{A,ij} - b - \pi\right) d\rho_{A,ij}\times d\rho\right) \\
& + \frac{a_c a}{2(1 + \|w_c+w\|^2)}\cos\left((w_c+w)^\top x + b_c + b\right) d\rho_c\times d\rho \\
& + \frac{a_c a}{2(1 + \|w_c-w\|^2)}\cos\left((w_c-w)^\top x + b_c - b\right) d\rho_c\times d\rho \\
& - \frac{a_f}{1 + \|w_f\|^2}\cos(w_f^\top x + b_f)d\rho_f.
\end{aligned}
$$

It is straightforward to verify that $v_t, \tilde{v}_t \in L^\infty(\mathbb{R}^n)$ with $(I - \Delta)\tilde{v}_t = v_t$. Note that the PDE $(I - \Delta)u = v_t$ has a unique solution in $\mathcal{S}_d'$, the space of tempered distributions, since the solution $u$ can be expressed in terms of the inverse Fourier transform of $v_t$, i.e. $u = (I - \Delta)^{-1}v_t = \mathcal{F}^{-1}\left(\frac{1}{1+\|\xi\|^2}\mathcal{F}v_t\right)$. Therefore the uniqueness implies that $\tilde{v}_t = (I - \Delta)^{-1}v_t$.

According to the support of $\rho_{A,ij}$, we only need to consider $w_{A,ij}$ with $\|w_{A,ij}\| \leq R_A$. For any $w$, if $\|w\| \geq 2R_A$, then

$$\left| \frac{a_{A,ij}a\langle w_{A,ij}, e_i\rangle\langle w, e_j\rangle}{2(1 + \|w_{A,ij} \pm w\|^2)} \right| \leq \left| \frac{a_{\partial A,ij}a\langle w_{A,ij}, e_i\rangle\langle w, e_j\rangle}{\frac{1}{2}\|w\|^2} \right| \leq |a_{A,ij}a|.$$

On the contrary, if $\|w\| < 2R_A$, then

$$\left| \frac{a_{A,ij}a\langle w_{A,ij}, e_i\rangle\langle w, e_j\rangle}{2(1 + \|w_{A,ij} \pm w\|^2)} \right| \leq \frac{2R_A^2}{2}|a_{A,ij}a| = R_A^2|a_{A,ij}a|.$$

Combining the above two cases, we obtain that

$$\left| \frac{a_{A,ij}a\langle w_{A,ij}, e_i\rangle\langle w, e_j\rangle}{2(1 + \|w_{A,ij} \pm w\|^2)} \right| \leq \max\{R_A^2, 1\} \cdot |a_{A,ij}a|, \quad \forall\, w \in \mathbb{R}^d,\ w_{A,ij} \in \overline{B}_{R_A}^d. \tag{D.1}$$

Similarly, we also have that

$$\left| \frac{a_{A,ij}a\langle w, e_i\rangle\langle w, e_j\rangle}{2(1 + \|w_{A,ij} \pm w\|^2)} \right| \leq 2\max\{R_A^2, 1\} \cdot |a_{A,ij}a|, \quad \forall\, w \in \mathbb{R}^d,\ w_{A,ij} \in \overline{B}_{R_A}^d. \tag{D.2}$$

Using (D.1), (D.2), and Lemma 3.3 (i), we can estimate the Barron norm of $\tilde{v}_t = (I - \Delta)^{-1}v_t$ as follows

$$\begin{aligned}
\left\|(I - \Delta)^{-1}v_t\right\|_{\mathcal{B}_{R_{\tilde{v},t}}^1(\mathbb{R}^d)} \leq{}& 6d^2\max\{R_A^2, 1\}\int |a_{A,ij}|d\rho_{A,ij}\int |a|d\rho \\
& + \int |a_c|d\rho_c \int |a|d\rho + \int |a_f|d\rho_f \\
\leq{}& 6d^2\max\{R_A^2, 1\}(\ell_A + \epsilon)\left(\|u_t\|_{\mathcal{B}_{R_{u,t}}^1(\mathbb{R}^d)} + \epsilon\right) \\
& + (\ell_c + \epsilon)\left(\|u_t\|_{\mathcal{B}_{R_{u,t}}^1(\mathbb{R}^d)} + \epsilon\right) + (\ell_f + \epsilon),
\end{aligned}$$

where $R_{\tilde{v},t} = R_{u,t} + \max\{R_A, R_c, R_f\}$. The estimate above directly implies that

$$\left\|(I - \Delta)^{-1}v_t\right\|_{\mathcal{B}_{R_{\tilde{v},t}}^1(\mathbb{R}^d)} \leq \left(6\ell_A\max\{R_A^2, 1\}d^2 + \ell_c\right)\|u_t\|_{\mathcal{B}_{R_{u,t}}^1(\mathbb{R}^d)} + \ell_f.$$

Then we can get (3.7) by applying Lemma 3.3 (i)-(ii). $\qquad\square$