# OpenReview forum: "On the Representation of Solutions to Elliptic PDEs in Barron Spaces"
_NeurIPS.cc/2021/Conference — NeurIPS 2021 Spotlight_

### Official Review · Reviewer_1yLq · 2021-07-05

**Rating:** 9
**Confidence:** 4

**Summary:**

This paper studies the representation of solutions of second order elliptic PDEs using two layer neural networks,
and gives sufficient conditions so that this can be done without curse of dimension (the number of units grows at
most polynomially in the dimension).  Precise theorems to this effect are stated and proven.
The mathematical framework includes the definition of a Barron space of functions (essentially, finite norm of the
final layer weights), many structural results about this function space, and an iterative solution scheme with its analysis.


**Limitations And Societal Impact:**

No apparent direct societal impact.

**Main Review:**

In recent years there has been much interest in developing numerical methods which take advantage of machine
learning concepts and technology, including the use of feedforward neural networks (FFNs) to represent solutions of PDEs.
There is a fair amount of numerical evidence that FFNs mitigate the curse of dimension for high dimensional PDEs of
practical interest, such as the HJB equation and equations of mathematical finance.
The present work studies the representation of solutions of second order elliptic PDEs using two layer neural networks,
and gives sufficient conditions so that this can be done without curse of dimension, here meaning the number of units grows at
most polynomially in the dimension.
The arguments do not use special properties of the PDEs and thus are more general than most previous work.
Their main conceptual tool is the definition of a Barron space of functions -- essentially, functions with a two layer FNN representation of arbitrary width such that the final layer weights have finite norm.
This definition is inspired by 1993 work of Barron (ref. [2]) as developed in ref. [8].
The authors develop the idea systematically in order to implement an iterative solution scheme and its analysis,
thus proving their theorems 2.9 and 2.10 which are precise forms of the claims.

The authors adequately cite the main works in this area that I know of.
The previous work closest to theirs is Marwah et al ref. [30], which uses a very similar iterative scheme to get similar results.
However ref. [30] does not define and systematize the function spaces involved as clearly.
I found the arguments in the present paper somewhat simpler and clearer, also they have some advantages (listed in "related works").   Also ref. [30] is not yet published and the preprint only appeared in March 2021, so the citation to it here seems adequate.

**Time Spent Reviewing:**

1

---

> ### Author Response · Authors · 2021-08-10
> **Response to Reviewer 1yLq**
>
> Thanks a lot for your very encouraging comments and feedback. The comparison with Ref. [30] is further addressed in the reply to Reviewer oNuF.

---

> > ### Comment · Reviewer_1yLq · 2021-09-10
> > **Thanks**
> >
> > Thanks for your reply.

---

### Official Review · Reviewer_oNuF · 2021-07-16

**Rating:** 6
**Confidence:** 3

**Summary:**

The paper derives complexity estimates of the solutions of second-order linear elliptic PDEs in the Barron space. It proves the solution of the PDE is ε-close with respect to the H1 norm to a Barron function and the complexity depends polynomially on the dimension d. The theorem implies physics-informed neural networks can efficiently approximate the solution of second-order linear elliptic PDEs.

**Limitations And Societal Impact:**

The second-order linear elliptic equation are in general easy for conventional solvers. The higher dimension cases are not very common to my opinion. It will be very interesting if the work can be extended to nonlinear PDE so that it can be applied to equations such as Navier-Stokes.

**Main Review:**

Compared to previous work, the paper focuses on deriving complexity estimates of the solution in the integral-representation-based Barron space.

Pros:
(1) the paper uses the Barron spaces techniques to improve the previous theorem.
(2) the writing is clear.
(3) clean related works.

Cons:
(1) the improvement looks subtle compared to previous work, especially [30], which also proves complexity estimates for elliptic equations. (2) The relief of assumption from 2 layer activations to 1 layer seems not very significant.

I have to admit that I am not very confident to assess the theoretical contribution and proof techniques. But overall, I like this work. I would vote for the border of acceptance.

**Time Spent Reviewing:**

2

---

> ### Author Response · Authors · 2021-08-10
> **Response to Reviewer oNuF**
>
> Thank you very much for the constructive comments and suggestions. We will revise the manuscript to further clarify these points. Here are our responses:
>
> 1. In Cons (1), it was stated that our work only had subtle improvement from the work of Marwah et al [30]. In our opinion, compared with [30], our work has the following significant improvements:
>      + The result in [30] establishes error estimate in $L^2$ norm, while our result gives error estimate in $H^1$ norm, and thus is stronger and is also more useful for the study of second-order elliptic PDEs.
>      + We show that the solution of elliptic PDE can be approximated efficiently by a simple neural network, say a two-layer neural network with a single activation function. As a comparison, the neural network used in [30] is potentially rather deep and also with multiple activation functions. We note that the referee mentioned that "from 2 layer activations to 1 layer" in comparison of our result with [30]; we suspect this might be a misunderstanding of the results.
>       + The analysis in [30] needs an assumption that the source term of the PDE can be approximated by a linear combination of eigenfunctions of the elliptic operator corresponding to the first $k$ eigenvalues, which is hard to verify in practice. In addition, their final complexity estimate involve with the $\lambda_k$, the $k$-eigenvalue, which is implicit and cannot be bounded a priori; In fact, $\lambda_k$ can actually become very large for large $k$ as $\lim_{k\rightarrow \infty}\lambda_k=\infty$. In contrast, our work completely removes such assumption on the right-hand side and also dependence of the bounds on $\lambda_k$. This is achieved by employing the preconditioning technique in the proof.
>
>  2. It was stated that one can solve linear elliptic PDEs easily using traditional methods. This is true of course for low-dimensional cases. However, the main interest of using neural networks is to solve elliptic PDEs in high dimension, such that the complexity of conventional methods will grow exponentially in the dimension $d$. Thus, one of the main focuses of our analysis is to obtain error estimates with milder dependence on the dimension.
>
>  3. It was questioned about the prevalence of high-dimensional linear elliptic PDEs. In fact, such equations are quite relevant and fundamental to science and engineering: Perhaps the most famous example is the governing equation of (static) many-body quantum systems, i.e., the Schrödinger equation. Another example is the (static) Fokker-Planck equation (i.e., Kolmogorov equations) which play important roles in several branches of science and engineering.
>
>  4. About generalization to nonlinear PDEs, we agree that extending the results to nonlinear PDEs is certainly one of the interesting future directions. Our result can be useful for certain nonlinear PDEs when the nonlinearity is weak so that one can employ an iterative scheme by linearization. Extending the results to general nonlinear PDEs is extremely challenging, and certainly beyond the scope of our work. It is left for future investigations.
>   The Navier-Stokes equation, while interesting, is not really the focus of neural-network-based methods, since it is a low-dimensional equation and has been studied extensively using conventional numerical methods for PDEs.

---

> > ### Comment · Reviewer_oNuF · 2021-08-24
> > **Concerns addressed**
> >
> > Thank the author for the informative response. My concerns are resolved.

---

### Official Review · Reviewer_b2vX · 2021-07-25

**Rating:** 7
**Confidence:** 3

**Summary:**

This paper studies the neural network approximation of high-dimensional elliptic PDEs. Specifically, it shows that when the source term and the coefficient functions lie in a Barron space, the solution can be approximated by two-layer neural networks without the curse of dimensionality.  The Barron norm characterizes the approximability of functions by two-layer neural networks.  The primary technique is to represent the PDE solution as the fixed point of the preconditioned steepest descent iteration.  Then,  it shows that the iteration satisfies two properties: (1) The iteration converges to the fixed point exponentially fast; (2) Each iteration only amplifies the Barron norm by a small fact.  To guarantee the first property, the PDE must be strongly elliptic. To guarantee the second property, the Barron space must be closed under multiplication, which excludes the most commonly-used activation functions.

**Limitations And Societal Impact:**

See the main review.

**Main Review:**

## Quality and clarity
This paper is well-written, and I can follow the authors' ideas very smoothly.  I went through the proof and they seem correct to me, though there are many tiny (no essential) errors.

## Originality
The result of this paper is obtained from a novel combination of the Barron space theory of two-layer neural networks and the property of elliptic PDEs, in particular representing the PDE solution as the fixed point of the preconditioned steepest iteration.  The authors also develop some interesting extensions of Barron space to bounded smooth activation functions, which may be of independent interest.
Given the previous work, the main result (Theorem 2.10) is not surprising to me.  However,  I believe that the rigorous proof is not straightforward, which can also be seen from this paper, which extensively exploits the structures of the two-layer neural nets and the strongly elliptic PDE.  Therefore, in my opinion, this work makes a decent contribution to understanding high-dimensional PDEs and the neural net approximation.

## Limitations and questions
- Assumption 2.7 excludes all the commonly-used activation functions.  I understand that this assumption is needed to establish the closeness of Barron space under multiplication. But, can authors comment more on this point? For instance, is it essential or just a technical issue?

- The main result 2.10 seems to suggest that cosine activation gives rise to a much better rate. Does the cosine activation function work better in practice under the setting of the theorem?


# Minor issues
- In Theorem 2.10, the upper bounds should depend on $m(\Omega)$  instead of $\sqrt{m(\Omega)}$.
- In line 232, the upper bound of $k$ at least has a factor $1/\varepsilon^2$, which comes from the Monte-Carlo discretization.
- In Proposition A.1 and the proof, $R_f$ is not defined.


**Time Spent Reviewing:**

10

---

> ### Author Response · Authors · 2021-08-10
> **Response to Reviewer b2vX**
>
> Thanks a lot for your helpful and insightful comments and questions. We will go through the manuscript carefully to correct the minor mistakes and typos. Here are our responses:
>
>  1. About Assumption 2.7 on activation functions, we agree with the referee that Assumption 2.7 is slightly restrictive. As the referee pointed out, it is an essential assumption to guarantee the closedness of multiplication in Barron spaces corresponding to two-layer networks.  We expect Assumption 2.7 may be weakened or removed if we allow for approximation with multi-layer neural networks. We will make a comment on this in the revision.
>
>  2. It was asked whether cosine worked better than other activation functions as suggested by Theorem 2.10. In some cases, periodic activation functions have shown better empirical performance in practice, see e.g. the work of Sitzmann et al [42].
>
>  3. Thanks for catching the mistake that in the statement of Theorem 2.10, it should be $m(\Omega)$ instead of  $\sqrt{m(\Omega)}$. We will fix that.
>
>  4. It was commented that in line 232, $1/\epsilon^2$ should be included in the upper bound of $k$. In fact, it is already included in the current version since $\mathcal{O}(d^{\beta'|\ln \epsilon|})=\mathcal{O}\left((\frac{1}{\epsilon})^{\beta' \ln d}\right)$.
>
> 5. It was commented that we did not define $R_f$ in Proposition A.1 and its proof. We introduce $R_f$ in line 529, which is perhaps easy to miss. We will make this more clear in the revision.

---

### Official Review · Reviewer_M9Yp · 2021-07-30

**Rating:** 9
**Confidence:** 3

**Summary:**

The objective of this paper is to prove that the solution $u$ of a certain class of PDEs, of the form
\begin{equation}\label{eq:PDE}
- \nabla\cdot (A \cdot \nabla u) + cu = f \mbox{ on } \mathbb{R}^d
\end{equation}
for some matrices $A$, coefficient $c$ and right-hand side $f$ can be well-approximated by functions belonging to some Barron spaces, provided that $A$, $c$ and $f$ belong to some Barron spaces as well. The precise approximation results are stated
in Theorems~2.9 and~2.10 of the manuscript.

Such kinds of approximation result are very interesting since they can give some insight on how well the solutions of some high-dimensional PDEs can be approximated by neural networks-based numerical methods. Indeed,
it is proved in earlier works that elements of these Barron spaces can be approximated in the $H^1$ norm by two-layer neural networks.

The article is organized as follows: notation and assumptions on the data of the problem are introduced in Section~2.1. The definition of Barron spaces is recalled in Section~2.2, together with approximation results in the $H^1$ norm of elements of Barron spaces
by two-layer neural networks (Theorem~2.5). The proofs of these results is postponed in Appendix~B. The main results of the paper concerning the approximation of the solution of PDE (\ref{eq:PDE}) by elements of a Barron space are stated in Theorem~2.9 and Theorem~2.10.
The main idea of the proofs of these results is sketched in Section~3, together with the statement of useful auxiliary lemmas. The detailed proofs of these results are given in the Appendix.

**Limitations And Societal Impact:**

No potential negative societal impact identified

**Main Review:**

The article is very well written and the presented results are very interesting with regards to the theoretical questions that neural-networks based numerical methods for the resolution of high-dimensional PDEs are rising. The proofs of the results all seem
correct to me although I did not check all the details. Hence, I am in favor of the publication of the present manuscript in NeurIPS, up to a few minor modifications highlighted below.

\begin{itemize}
 \item p.4: Def.~2.2: precise the definition of $\overline{B}_R^d$ l. 153
 \item Theorem~2.5, p. 5, l. 184: In this theorem $g$ is assumed to belong to the space $\mathcal B_R^2(\Omega)$. However, in Proposition~2.4, it is stated that $\mathcal B_R^\infty(\Omega) = \mathcal B_R^p(\Omega) = \mathcal B_R^1(\Omega)$ for $1\leq p \leq \infty$.
 The use of the exponent $2$ in Theorem~2.5 is then confusing. I suspect a typo and that the author(s) actually meant to assume that $g$ belongs to $\mathcal B_R^1(\Omega)$. Correct? Similar remark/question at l.205 of the same page (Assumptions~2.8) where
 $l_{d,2}$ is defined as $\|\sigma''\|_{\mathcal B^2_{R_{d,2}}}$. Do the authors mean $l_{d,2}:= \|\sigma''\|_{\mathcal B^1_{R_{d,2}}}$?  Please check in the manuscript if other similar typos are present.
 \item p.5, l. 200: Do the authors simply mean by ${\rm im}\sigma \neq \{0 \}$ the fact that $\sigma$ has to be non identically equal to $0$? I am unsure of the meaning of the ${\rm im}$ notation there.
 \item Theorem~2.9, l.214, p. 6: Could the authors comment on the fact that the dimension $d$ has to be assumed greater than $3$ for the result to hold? Could the authors add a remark on the reason why wuch a result could not hold in lower dimension?
\end{itemize}


**Time Spent Reviewing:**

5

---

> ### Author Response · Authors · 2021-08-10
> **Response to Reviewer M9Yp**
>
> Thank you very much for your very encouraging comments and feedback. Here are our responses to the minor issues you pointed out:
>
> 1. About the definition of $\bar{B}^d_R$ in Definition 2.2, in the current version, we define it in the notation part in Section 2.1. We will make it more clear and recall it in Definition 2.2 in the revision.
>
> 2. It was suggested that we should use $\mathcal{B}^1_R(\Omega)$,  instead of $\mathcal{B}^2_R(\Omega)$, in the statement of Theorem 2.5 and Assumption 2.8. Indeed the statement in Assumption 2.8 was a typo and it should be $\mathcal{B}^1_R(\Omega)$. We will correct this and also go over the manuscript carefully in the revision for other possible minor mistakes.
>   As for Theorem 2.5, previously we assumed $g\in\mathcal{B}^2_R(\Omega)$, rather than $g\in\mathcal{B}^1_R(\Omega)$, as we used $g\in\mathcal{B}^2_R(\Omega)$ in the proof. As the referee pointed out, however, these two assumptions are equivalent due to Proposition 2.4. We will change $g\in\mathcal{B}^2_R(\Omega)$ to $g\in\mathcal{B}^1_R(\Omega)$ to make it more consistent with the Barron norm we estimate in the iterative scheme.
>
> 3. The referee is correct that $\text{im}(\sigma)\neq\{0\}$ means that $\sigma$ takes some non-zero values. We will state this more explicitly in the revised version.
>
> 4. It was suggested to remark on the assumption $d\geq 3$ in Theorem 2.9. We assumed $d\geq 3$ in Theorem 2.9 since its proof relies on Lemma 3.4 which is valid under the same assumption. Specifically, the proof of Lemma 3.4 further relies on an upper bound of the $L^1$-norm of the Green's function of the $d$-dimensional screened Laplacian operator that is derived after calculating an analytic formula of such Green's function for $d\geq 3$ in the work by  E and Wojtowytsch [9]. In fact, we realized that the result for $d = 1$ or $2$ also held with a slightly different proof. We will include this in the revision and add some clarification.

---

### Author Response · Authors · 2021-08-10
**Sincere Thanks**

We appreciate the reviewers for spending their time carefully read our manuscript and for their helpful and encouraging comments. We have responded to their detailed comments in separate replies. Hopefully, our replies have addressed all the concerns.

---

### Decision · Program_Chairs · 2021-09-27

**Decision:**

Accept (Spotlight)

**Comment:**

All the reviewers agree that it is a very interesting paper in its field. I share the same opinion.

The main contribution to this paper is to show that solutions of elliptic PDE of the form
\begin{equation}
\- \nabla \cdot (A \nabla u) + cu = f
\end{equation}
can be approximated by a two-layer neural network in some Sobolev norm. This kind of results is really important due to the increase popularity of PDE solvers based on deep neural networks.

The main assumption considered by the authors is that the coefficients $A,c$ and $f$ all belong to some Baron spaces which have to be closed by multiplication. To obtain such a results, the authors builds upon results on Baron spaces recently obtained in the literature and on the representation of solutions of elliptic PDE as fixed-points of a steepest descent scheme.

While, this result and the strategy of the proof may seem not really surprising, the paper is very well-written and succeed in gathering all the ingredients of the proofs smoothly.

That being said, some points have been raised by the reviewers. I encourage the authors to address these in the final version of their paper.